# Standardized mean differences cause funnel plot distortion in publication bias assessments

Peter-Paul Zwetsloot[1,2]*, Mira Van Der Naald[1,2], Emily S Sena[3], David W Howells[4], Joanna IntHout[5], Joris AH De Groot[6], Steven AJ Chamuleau[1,2,7], Malcolm R MacLeod[3], Kimberley E Wever[8]*

[1]Cardiology, Experimental Cardiology Laboratory, University Medical Center Utrecht, Utrecht, Netherlands; [2]Netherlands Heart Institute, Utrecht, Netherlands; [3]Center for Clinical Brain Sciences, University of Edinburgh, Edinburgh, United Kingdom; [4]School of Medicine, University of Tasmania, Hobart, Australia; [5]Department for Health Evidence, Radboud Institute for Health Sciences, Radboud University Medical Center, Nijmegen, Netherlands; [6]Julius Center for Health Sciences and Primary care, University Medical Center Utrecht, Utrecht, Netherlands; [7]Regenerative Medicine Center, University Medical Center Utrecht, Utrecht, Netherlands; [8]Systematic Review Centre for Laboratory Animal Experimentation (SYRCLE), Radboud Institute for Health Sciences, Radboud University Medical Center, Nijmegen, Netherlands

*For correspondence:
P.P.M.Zwetsloot@umcutrecht.nl
(P-PZ);
kim.wever@radboudumc.nl (KEW)

**Competing interests:** The authors declare that no competing interests exist.

**Abstract** Meta-analyses are increasingly used for synthesis of evidence from biomedical research, and often include an assessment of publication bias based on visual or analytical detection of asymmetry in funnel plots. We studied the influence of different normalisation approaches, sample size and intervention effects on funnel plot asymmetry, using empirical datasets and illustrative simulations. We found that funnel plots of the Standardized Mean Difference (SMD) plotted against the standard error (SE) are susceptible to distortion, leading to overestimation of the existence and extent of publication bias. Distortion was more severe when the primary studies had a small sample size and when an intervention effect was present. We show that using the Normalised Mean Difference measure as effect size (when possible), or plotting the SMD against a sample size-based precision estimate, are more reliable alternatives. We conclude that funnel plots using the SMD in combination with the SE are unsuitable for publication bias assessments and can lead to false-positive results.

DOI: https://doi.org/10.7554/eLife.24260.001

## Introduction

Systematic reviews are literature reviews intended to answer a particular research question by identifying, appraising and synthesizing all research evidence relevant to that question. They may include a meta-analysis, a statistical approach in which outcome data from individual studies are combined, which can be used to estimate the direction and magnitude of any underlying intervention effect, and to explore sources of between-study heterogeneity. Simultaneously, meta-analysis can be used to assess the risk of publication bias: the phenomenon that published research is more likely to have positive or statistically significant results than unpublished experiments (*Dwan et al., 2013*). Meta-analyses are routinely used in clinical research to guide clinical practice and healthcare policy, reduce research waste and increase patient safety (*Chalmers et al., 2014*). The use of meta-analysis

continues to increase (*Bastian et al., 2010*) and it has become more common to apply these approaches to the synthesis of preclinical evidence (*Korevaar et al., 2011*). Importantly, preclinical studies are, generally, individually small, with large numbers of studies included in meta-analysis, and large observed effects of interventions. This contrasts with clinical research, where meta-analyses usually involve a smaller number of individually larger experiments with smaller intervention effects.

This calls for methodological research to ascertain whether approaches to data analysis routinely used in the clinical domain are appropriate in the pre-clinical domain and for resources that guide and inform researchers, reviewers and readers on best practice. In this light, we present findings which show that the use of the standardized mean difference (SMD) measure of effect size in funnel plots can introduce a risk of incorrect assessment of publication bias, particularly in meta-analyses of preclinical data characterised by a large number of individually small studies with large observed effects.

## Formulation of raw mean difference, standardized mean difference and normalized mean difference

To combine data statistically on *e.g.* the effects of an intervention which has been tested in several studies, outcome measures first need to be expressed on a common scale. Such scales include (for binary outcomes) the risk or odds ratios; and for continuous data a raw mean difference (RMD), SMD or normalized mean difference (NMD).

The RMD can be used when all outcome data are in the same measurement unit, and the interpretation of the outcome is the same in all settings (*i.e.* the reported measurement unit of the change in outcome has the same meaning in all studies). The RMD is calculated by subtracting the mean outcome value in the control group ($M_{ctrl}$) from the mean in the intervention group ($M_{int}$):

$$RMD = M_{int} - M_{ctrl}. \tag{1}$$

The observed standard deviation (SD) is likely to differ between experimental groups, and therefore the standard error (SE) of the RMD is calculated as:

$$SE_{RMD} = \sqrt{\frac{SD_{int}^2}{n_{int}} + \frac{SD_{ctrl}^2}{n_{ctrl}}}, \tag{2}$$

where n is the sample size per group.

In cases where the measurement unit, or the interpretation of the outcome, or both differ between studies (*e.g.* a given change in infarct size measured in $mm^3$ has a different consequence in the mouse brain than in the rat brain), the intervention effect may be expressed as an SMD. For each study the SMD is obtained by dividing the RMD by that study's pooled standard deviation ($SD_{pooled}$) to create an effect estimate that is comparable across studies:

$$SMD = d = \frac{M_{int} - M_{ctrl}}{SD_{pooled}} \tag{3}$$

, where $SD_{pooled}$ is:

$$SD_{pooled} = \sqrt{\frac{(n_{ctrl} - 1)\,SD_{ctrl}^2 + (n_{int} - 1)\,SD_{int}^2}{n_{ctrl} + n_{int} - 2}} \tag{4}$$

Thus, the SMD expresses the intervention effect in all studies in the same new unit: the SD.

For each study, the standard error (SE) of the SMD can be approximated using the sample sizes (n) and the effect estimate (SMD):

$$SE_{SMD} = \sqrt{\frac{(n_{ctrl} + n_{int})}{n_{ctrl} * n_{int}} + \frac{SMD^2}{2 * (n_{ctrl} + n_{int})}} \tag{5}$$

Of note, *Equations 3 and 5* estimate the SMD using the approach of Cohen (*Cohen, 1988*); this estimate is therefore termed Cohen's *d*. However, Cohen's *d* tends to overestimate the 'true' SMD and its variance when the sample sizes in the primary studies are small (*e.g.* <10). This bias can be

corrected using the approach of Hedges (*Hedges, 1981*), which adjusts both the SMD estimate and its variance by a correction factor based on the total sample size. The resulting estimate is the unbiased SMD known as Hedges' *g* (see *Supplementary file 2* for full equations). In many clinical meta-analyses, Hedges' *g* will be almost identical to Cohen's *d*, but the difference between the estimates can be larger in preclinical meta-analyses, where small sample sizes are more common.

A third effect measure commonly used for continuous data in preclinical meta-analyses is the normalised mean difference (NMD), which relates the magnitude of effect in the intervention group to that seen in untreated animals, with reference to the outcome in a normal, healthy animal (*Vesterinen et al., 2014*). A condition for using the NMD is that the baseline measurement in an untreated, unlesioned 'sham' animal is known, or can be inferred. For each study, the NMD is calculated as:

$$NMD = 100\% \times \frac{(M_{int} - M_{sham}) - (M_{ctrl} - M_{sham})}{(M_{ctrl} - M_{sham})} \tag{6}$$

where $M_{sham}$ is the mean score for normal, unlesioned and untreated subjects. The corresponding SE is calculated as:

$$SE_{NMD} = \sqrt{\frac{\left(100 * \frac{SD_{ctrl}}{M_{ctrl} - M_{sham}}\right)^2}{n_{ctrl}} + \frac{\left(100 * \frac{SD_{int}}{M_{ctrl} - M_{sham}}\right)^2}{n_{int}}} \tag{7}$$

(see *Supplementary file 2* for additional equations and (*Vesterinen et al., 2014*) for a comprehensive overview of (preclinical) meta-analysis methodology).

Note that *Equation 5* dictates that the $SE_{SMD}$ is correlated to the SMD effect size, whereas the SEs of the RMD (*Equation 2*) and NMD (*Equation 7*) are independent of the corresponding effect sizes.

## Funnel plots and publication bias

Funnel plots are scatter plots of the effect sizes of the included studies *versus* a measure of their precision, usually the SE or 1/SE. In the absence of bias and heterogeneity, funnel plots should be funnel-shaped and symmetrically centred around the summary effect estimate of the analysis, since 1) imprecise (smaller) studies will deviate further from the summary effect compared to precise (larger) studies and 2) studies are equally likely to overestimate or underestimate the true effect (*Figure 1A*). Assessment of the possible presence of publication bias frequently relies on a visual or analytical evaluation of funnel plot asymmetry. If studies showing small, neutral or controversial effects are more likely to remain unpublished, publication bias may occur. As a result, the funnel plot will become asymmetrical, and the summary effect estimate will shift accordingly (*Figure 1B*). Importantly, there are other causes of asymmetry in funnel plots. For instance, the true effect size in smaller (and therefore less precise) studies may be genuinely different from that in large studies (for instance because the intensity of the intervention was higher in small studies). For this reason, funnel plot asymmetry is often referred to as a method to detect small study effects, rather than being a definitive test for publication bias (*Rothstein et al., 2005*). In addition, artefacts and chance may cause asymmetry (as shown *e.g.* in this study).

## Theoretical explanation of SMD funnel plot distortion

In a meta-analysis using the SMD as effect measure, in the absence of publication bias, observed SMDs in a funnel plot will be scattered around the true underlying SMD. However, the dependency of the $SE_{SMD}$ on the observed SMD will impact the appearance of the funnel plot. When we review the equation for the $SE_{SMD}$, (*Equation 5*) the first component on the right of the '=' sign reflects the variance of the difference between the two group means, rescaled into pooled standard deviation units. Consequently, in this first part only $n_{ctrl}$ and $n_{int}$ play a role. The second component includes the squared SMD, and reflects the variation in the within-groups standard deviation as measured by $SD_{pooled}$ (*Equation 4*).

If there is no intervention effect, the SMD (and the second component) will be zero, and the SE will therefore depend solely on the sample size (*Equation 5* and *Figure 2A*). If an intervention effect is present, the SE will increase, as the size of $SMD^2$ in the equation will increase. This is no problem

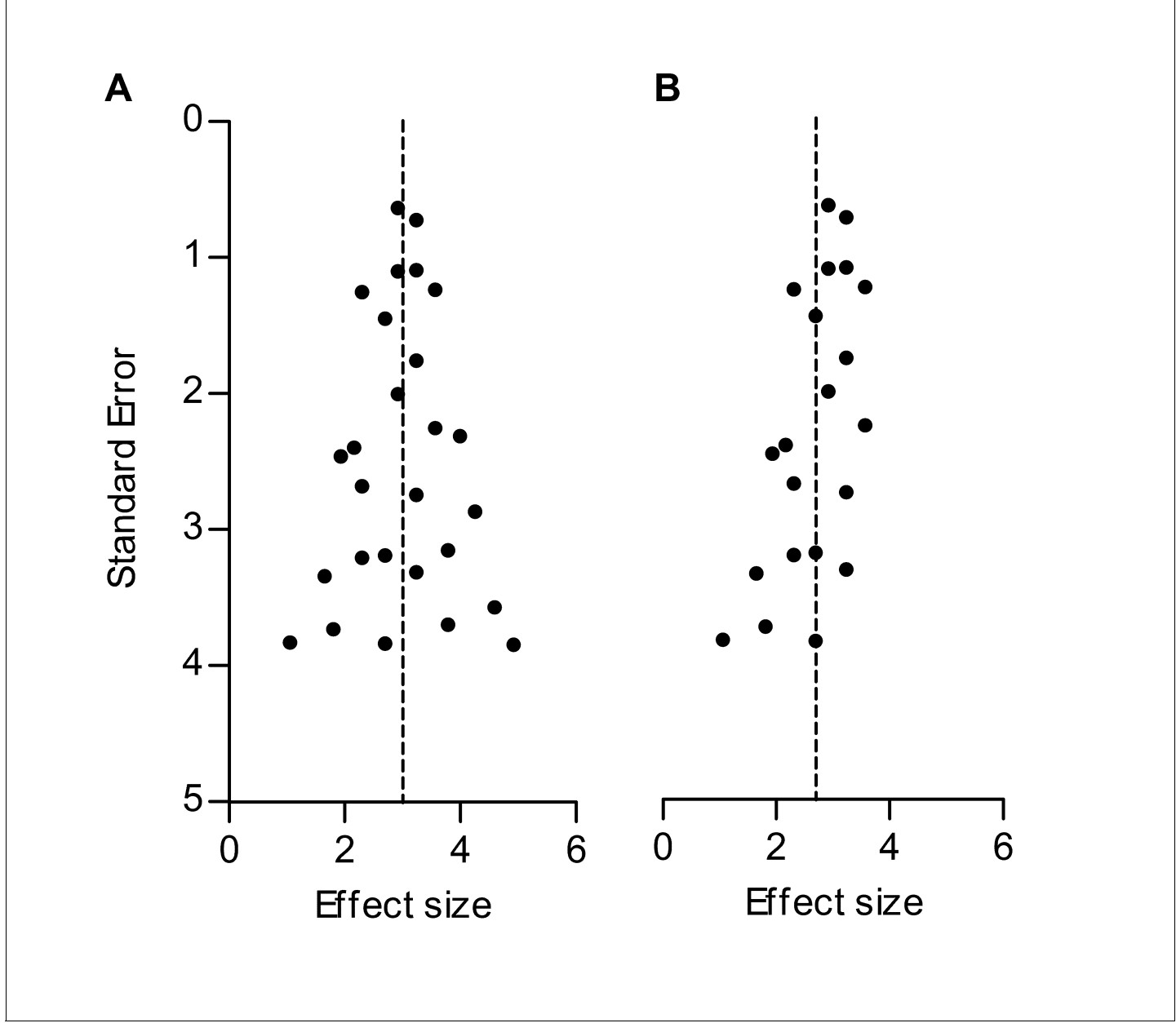

**Figure 1.** Hypothetical funnel plots in the absence (A) and presence (B) of bias. The precision estimate used is the standard error (SE). Dashed lines indicate the summary effect estimate.

DOI: https://doi.org/10.7554/eLife.24260.002

if the observed SMD is similar to the true SMD. However, a study with an observed SMD larger than the true SMD will have a larger SE. On the other hand, a study with an observed SMD smaller than the true SMD (but >0) will have a relatively small SE (*Figure 2B*). This will cause funnel plot distortion: studies with a relatively small effect size (and associated SE) will skew towards the upper left region of the plot, while studies with a relatively large effect size and SE will skew towards the bottom right region of the plot, as the associated SE of these studies will be relatively large. Because the SMD is squared in the equation for the SE, this holds true for both positive and negative SMDs (*Figure 2C*). The smaller the first component of *Equation 5*, the larger the influence of the SMD on the size of the SE, worsening the distortion when sample sizes are small. Of note, this component is

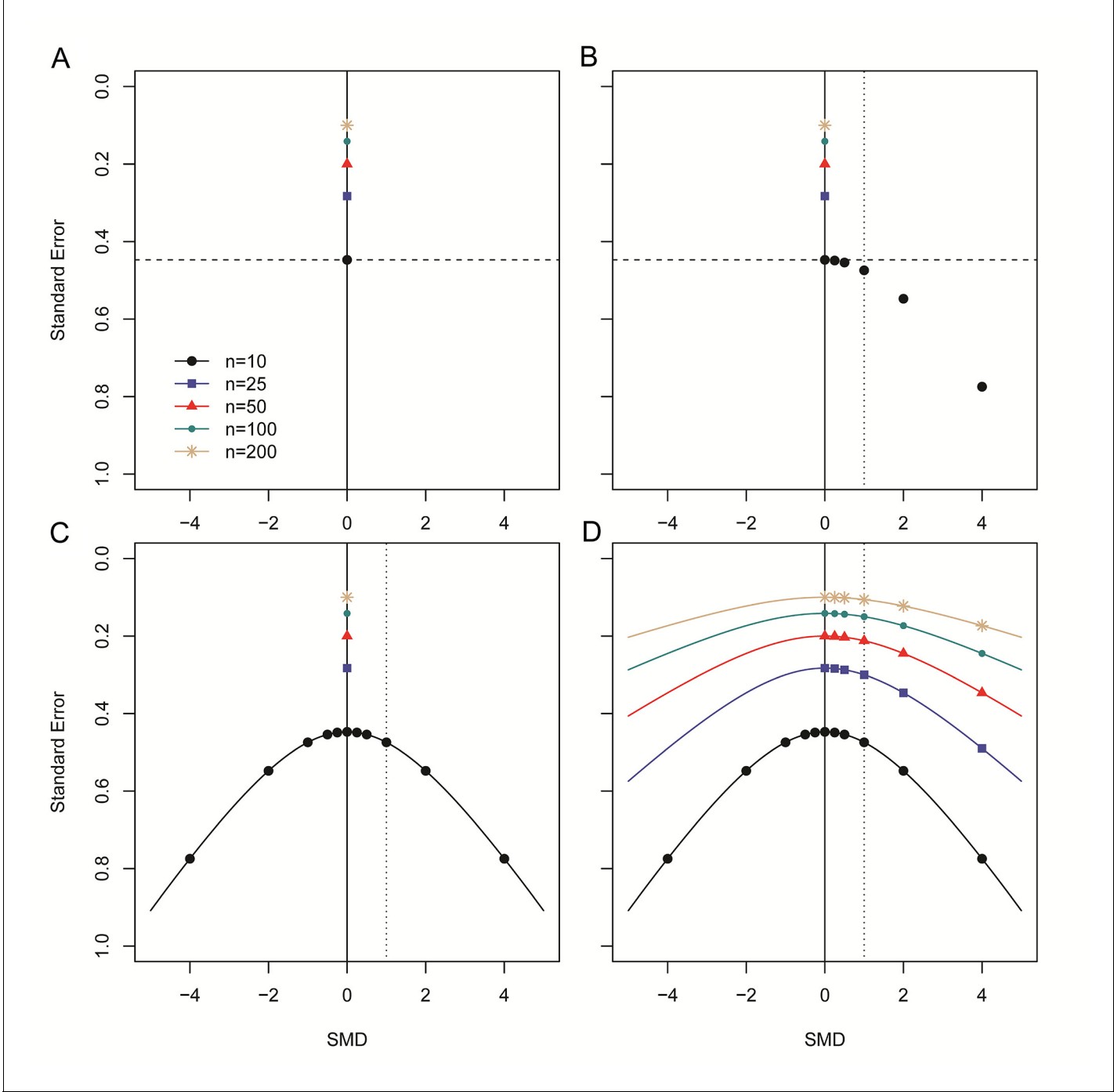

**Figure 2.** Step-wise illustration of distortion in SMD versus SE funnel plots. (**A**) Depicted are simulated studies with a sample size of respectively 10 (large black circles), 25 (blue squares), 50 (red triangles), 100 (small green circles) and 200 (gold asterisks) subjects per group, and an SMD of zero. The SE of these studies (indicated by the dashed line for studies with n = 10) solely depends on their sample size, as $SMD^2 = 0$ and therefore does not contribute to the equation for the SE. As expected, the SE decreases as the sample size increases. (**B**) Five data points from simulated studies with n = 10 and a stepwise increasing SMD are added to the plot. For these studies, the $SMD^2$ contributes to the equation for the SE, and the SE will decrease even though the sample size is constant. The dotted line represents a hypothetical summary effect of SMD = 1 in a meta-analysis. Note that when assessing a funnel plot for asymmetry around this axis, the data points with an SMD < 1 have skewed to the upper left-hand region, whereas studies with an SMD > 1 are in the lower right region of the plot. This distortion worsens as the SMD increases. (**C**) Because the SMD is squared in the equation for the SE, the same distortion pattern is observed for negative SMDs. Thus, funnel plots will be distorted most when the study samples sizes are small and SMDs are either very positive or very negative. (**D**) The same deviation is observed for simulated studies with larger sample sizes, however, the deviation decreases as the sample size increases, because the sample size will outweigh the effect of $SMD^2$ in the equation for the SE.
DOI: https://doi.org/10.7554/eLife.24260.003

smallest when group sizes are unequal. The effect of the second component on the SE, and the resulting distortion, is largest if the sample size is small and the SMD is large (*Figure 2D*).

In summary, a funnel plot using both the SMD and its SE may become asymmetrical in the absence of publication bias. When funnel plot distortion is assessed by visual inspection, this skewing might cause the plot to be interpreted as being asymmetrical and lead the observer to erroneously conclude that publication bias is present. Furthermore, funnel plot asymmetry is often tested statistically using Egger's regression (*Egger et al., 1997*) or Duval and Tweedie's trim and fill analysis (*Duval and Tweedie, 2000*), but neither of these analyses take the phenomenon described above into account, and their use may lead to erroneous conclusions that publication bias is present.

## Aim of this study

We investigated the reliability of RMD, SMD and NMD-based funnel plots for the assessment of publication bias in meta-analyses, using both empirical datasets and data simulations. We investigate the effect on the severity of funnel plot distortion of the study sample size, the number of studies in the meta-analysis and the magnitude of the intervention effect. We assess whether distortion can be avoided by using a precision estimate based on the sample size of the primary studies, as previously suggested for mean difference outcome measurements (*Sterne et al., 2011*). We then use this alternative approach to reanalyse published funnel plots, and show that these systematic reviews may have overestimated the severity of publication bias in their body of evidence. Our findings have important implications for the meta-research field, since authors may have reached incorrect conclusions regarding the existence of publication bias based on funnel plots using the SMD measure of effect size.

## Results

### Publication bias assessment using RMD *versus* SMD funnel plots of two preclinical RMD datasets

Dataset 1 (ischaemic preconditioning) contains 785 individual effect sizes (*Wever et al., 2015*). In the original analysis using the RMD as effect measure, funnel plot asymmetry was detected by Egger's regression ($p=1.7\times10^{-5}$), but no additional studies were imputed in trim and fill analysis (*Figure 3A*). When expressing the same data as SMD, funnel plot asymmetry increased substantially (*Figure 3B*; $p<1.0\times10^{-15}$, Egger regression) and 196 missing studies were imputed by trim and fill analysis, leading to adjustment of the estimated SMD effect size from 2.8 to 1.9.

Dataset 2 (stem cell treatments) contained 95 individual effect sizes (*Zwetsloot et al., 2016*). Funnel plot asymmetry was detected in the original analysis using RMD ($p=0.02$) and trim and fill analysis suggested a reduction in effect estimate of 0.1% after filling two additional studies (*Figure 3C*). In contrast, a funnel plot of the same data expressed as SMD showed asymmetry at a higher level of statistical significance ($p=3.4\times10^{-10}$, Egger regression), but no missing studies were imputed (*Figure 3D*).

### Data simulation results

Results of our first simulation (in the absence of publication bias) are shown in *Table 1*, and representative funnel plots of these simulations in *Figure 4* (small study sample size) and *Figure 4—figure supplement 1* (large study sample size). When we simulated no intervention effect, neither Egger's regression nor trim and fill analysis gave different results for the RMD vs. SE and SMD vs. SE analyses (*Table 1*, *Figure 4A,B,E and F* and *Figure 4—figure supplement 1*, panel A, B, E and F) and in ~95% of cases there was no evidence of asymmetry. Most simulated funnel plots were assessed as symmetrical, however, as expected, around 5% of the cases were considered asymmetrical by chance.

When we simulated the presence of an intervention effect ($\Delta\mu = 10$; RMD = 10 and SMD = 1 or $\Delta\mu = 5$; RMD = 5 and SMD = 0.5), again around 5% of the RMD funnel plot analyses were judged asymmetrical (*Table 1*, *Figure 4C and G*, and *Figure 4—figure supplement 1*, panel C and G). In contrast, when using the SMD, funnel plot asymmetry was detected in over 60% of the simulated funnel plots with $\Delta\mu = 10$, where the size of contributing studies was small (*Figure 4D and H* and *Figure 4—figure supplement 1*, panel D and H), increasing as the number of individual studies

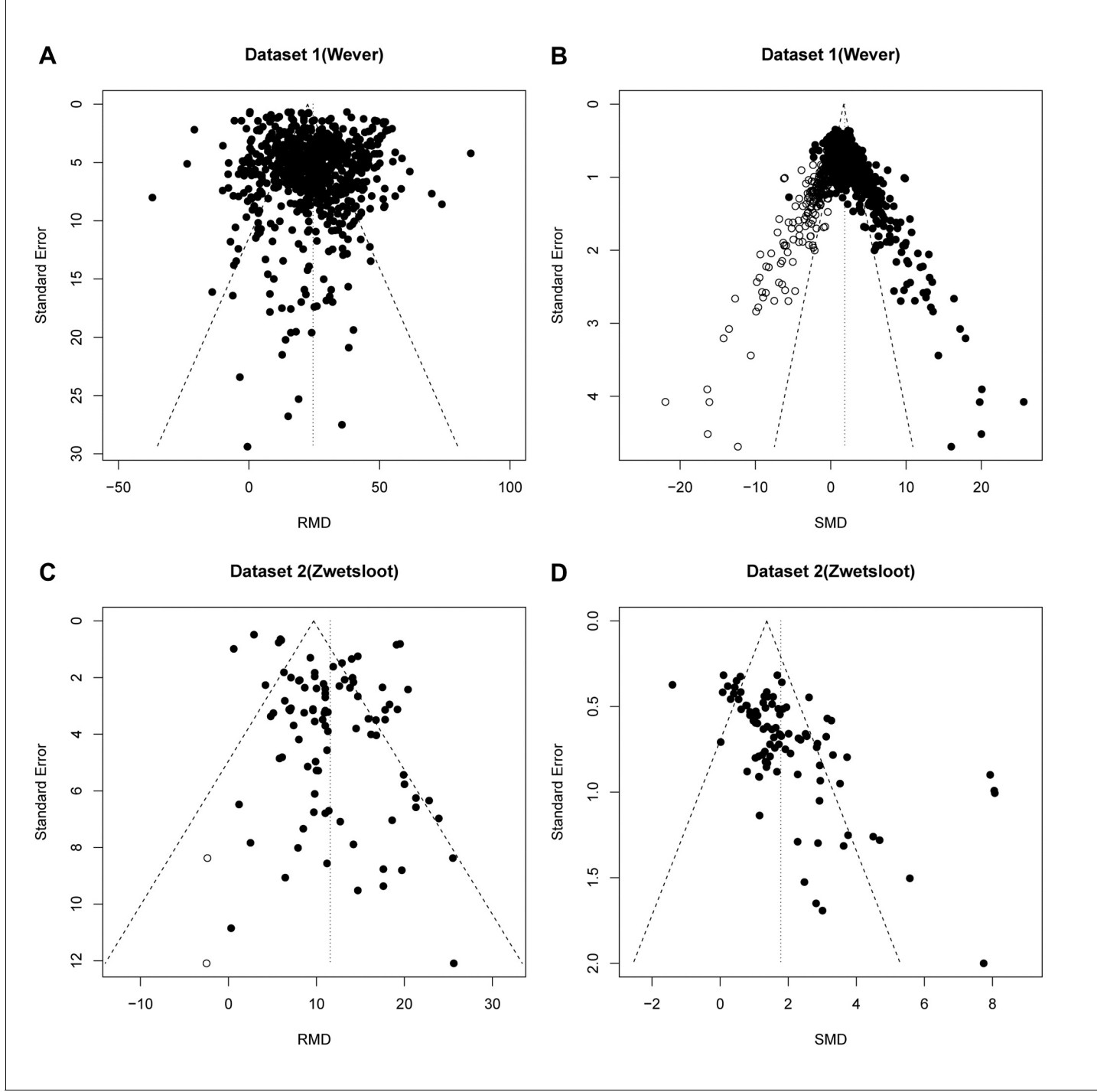

**Figure 3.** Reanalysis of data from Wever et al. (**A,B**) and Zwetsloot et al. (**C,D**), with funnel plots based on raw mean difference (RMD; **A,D**) or standardized mean difference (SMD; **B,D**). Filled circles = observed data points; open circles = missing data points as suggested by trim and fill analysis.

DOI: https://doi.org/10.7554/eLife.24260.004

contributing to the meta-analysis increased. When we modelled larger individual contributing studies (n = 60–320 subjects), respectively 9%, 34% and 100% of the SMD funnel plots with 30, 300 or 3000 studies were assessed as asymmetrical (*Table 1*, *Figure 4—figure supplement 1*). Trim and fill analysis resulted in on average 7% extra studies filled in preclinical simulation scenarios using the

**Table 1.** Study characteristics in relation to publication bias assessment in simulation of unbiased meta-analyses (simulation 1)

| Total study N | Δμ | No. of studies in MA | Effect measure | % of simulations with Egger's p<0.05 | No. of studies filled by T&F (mean(min - max)) | Overall effect size (mean(min - max)) | Overall effect size after T&F (mean(min - max)) |
|---|---|---|---|---|---|---|---|
| 12–30 | 0 | 30 | RMD | 6.2% | 2.1 (0–11) | 0.74(−12.2–11.3) | 0.0(−3.8–3.6) |
| | | | SMD(g) | 9.3% | 1.6 (0–10) | 0.1(−1.1–1.4) | 0.0(−0.36–0.33) |
| 12–30 | 5 | 30 | RMD | 4.9% | 2.1 (0–10) | 5.3(−3.4–19.1) | 5.0 (1.2–9.6) |
| | | | SMD(g) | 19.5% | 2.4 (0–10) | 0.55(−0.4–2.2) | 0.43 (0.11–0.74) |
| 12–30 | 10 | 30 | RMD | 4.6% | 2.0 (0–10) | 11.2 (1.2–20.4) | 10.0 (5.4–13.5) |
| | | | SMD(g) | 67.2% | 4.4 (0–10) | 1.16 (0.2–2.4) | 0.85 (0.5–1.2) |
| 12–30 | 0 | 300 | RMD | 4.8% | 25.4 (0–62) | 0.0(−15.2–12.3) | 0.0(−2.1–2.3) |
| | | | SMD(g) | 9.8% | 18.8 (0–57) | 0.0(−1.9–1.6) | 0.0(−0.2–0.2) |
| 12–30 | 5 | 300 | RMD | 5.5% | 25.1 (0–65) | 5.5(−10.2–23.7) | 5.0 (3.0–6.8) |
| | | | SMD(g) | 96.0% | 47.3 (0–70) | 0.55(−1.1–2.3) | 0.37 (0.28–0.50) |
| 12–30 | 10 | 300 | RMD | 5.9% | 25.8 (0–61) | 10.3(−11.1–29.0) | 10.0 (7.9–12.3) |
| | | | SMD(g) | 100% | 61.5 (40–76) | 1.0(−1.4–3.1) | 0.80 (0.70–0.89) |
| 12–30 | 0 | 3000 | RMD | 5.4% | 249 (0–453) | 0.0(−18.6–17.9) | 0.0(−1.4–1.3) |
| | | | SMD(g) | 8.7% | 175.1 (0–386) | 0.0(−2.1–2.6) | 0.0(−0.1–0.1) |
| 12–30 | 5 | 3000 | RMD | 4.4% | 252 (0–475) | 4.9(−13.0–21.1) | 5.0 (3.7–6.4) |
| | | | SMD(g) | 100% | 492(417 - 565) | 0.49(−1.7–2.9) | 0.36 (0.33–0.39) |
| 12–30 | 10 | 3000 | RMD | 5.0% | 250 (0–456) | 10.0(−7–27) | 10.0 (8.6–11.3) |
| | | | SMD(g) | 100% | 620(568 - 669) | 1.0(−0.7–4.5) | 0.79 (0.8–0.8) |
| 60–320 | 0 | 30 | RMD | 4.7% | 2.4 (0–10) | −0.2(−3.8–3.3) | 0.0(−1.3–1.3) |
| | | | SMD(g) | 5.0% | 2.4 (0–10) | 0.0(−0.4–0.4) | 0.0(−0.1–0.1) |
| 60–320 | 5 | 30 | RMD | 3.8% | 2.2 (0–10) | 4.8 (1.9–7.6) | 5.0 (3.8–6.1) |
| | | | SMD(g) | 5.2% | 2.4 (0–13) | 0.48 (0.2–0.8) | 0.5 (0.4–0.6) |
| 60–320 | 10 | 30 | RMD | 5.9% | 2.4 (0–10) | 10.0 (6.7–14.0) | 10.0 (8.7–11.2) |
| | | | SMD(g) | 7.9% | 2.6 (0–10) | 1.0 (0.6–1.3) | 1.0 (0.8–1.1) |
| 60–320 | 0 | 300 | RMD | 4.4% | 18.9 (0–58) | 0.1(−3.7–5.5) | 0.0(−0.5–0.6) |
| | | | SMD(g) | 4.6% | 17.3 (0–58) | 0.0(−0.4–0.5) | 0.0(−0.1–0.1) |
| 60–320 | 5 | 300 | RMD | 4.7% | 17.8 (0–63) | 4.9 (0.0–9.7) | 5.0 (4.4–5.6) |
| | | | SMD(g) | 11.8% | 20.7 (0–60) | 0.49 (0.0–0.9) | 0.49 (0.4–0.5) |
| 60–320 | 10 | 300 | RMD | 6.2% | 18.4 (0–63) | 10.1 (4.8–16.5) | 10.0 (9.4–10.6) |
| | | | SMD(g) | 33.9% | 29.5 (0–71) | 1.0 (0.5–1.7) | 0.97 (0.9–1.0) |
| 60–320 | 0 | 3000 | RMD | 5.3% | 140.0 (0–367) | 0.0(−6.5–5.6) | 0.0(−0.3–0.3) |
| | | | SMD(g) | 5.4% | 136.6 (0–348) | 0.0(−0.7–0.6) | 0.0 (0.0–0.0) |
| 60–320 | 5 | 3000 | RMD | 4.7% | 143 (0–331) | 5.0(−1.4–11.3) | 5.0 (4.7–5.3) |
| | | | SMD(g) | 69.0% | 243 (0–391) | 0.5(−0.1–1.2) | 0.48 (0.46–0.51) |
| 60–320 | 10 | 3000 | RMD | 5.0% | 135.8 (0–340) | 10.0 (4.6–16.2) | 10.0 (9.7–10.3) |
| | | | SMD(g) | 99.7% | 334.5(168–464) | 1.0 (0.47–1.61) | 0.97 (0.95–0.98) |

n = sample size; Δμ = difference in normally distributed means between intervention and control group; no. = number; MA = meta analysis; T and F = trim and fill analysis; RMD = raw mean difference; SMD(g)=Hedges' g standardized mean difference; SD = standard deviation

DOI: https://doi.org/10.7554/eLife.24260.008

RMD. Adjusting the overall effect estimate based on these filled data points improved the estimation of the simulated RMD in all scenarios. However, when using the SMD, the number of filled studies was much higher in many scenarios (up to 21% extra studies filled). As a result, the adjusted overall effect estimate after trim and fill in SMD funnel plots tended to be an underestimation of the

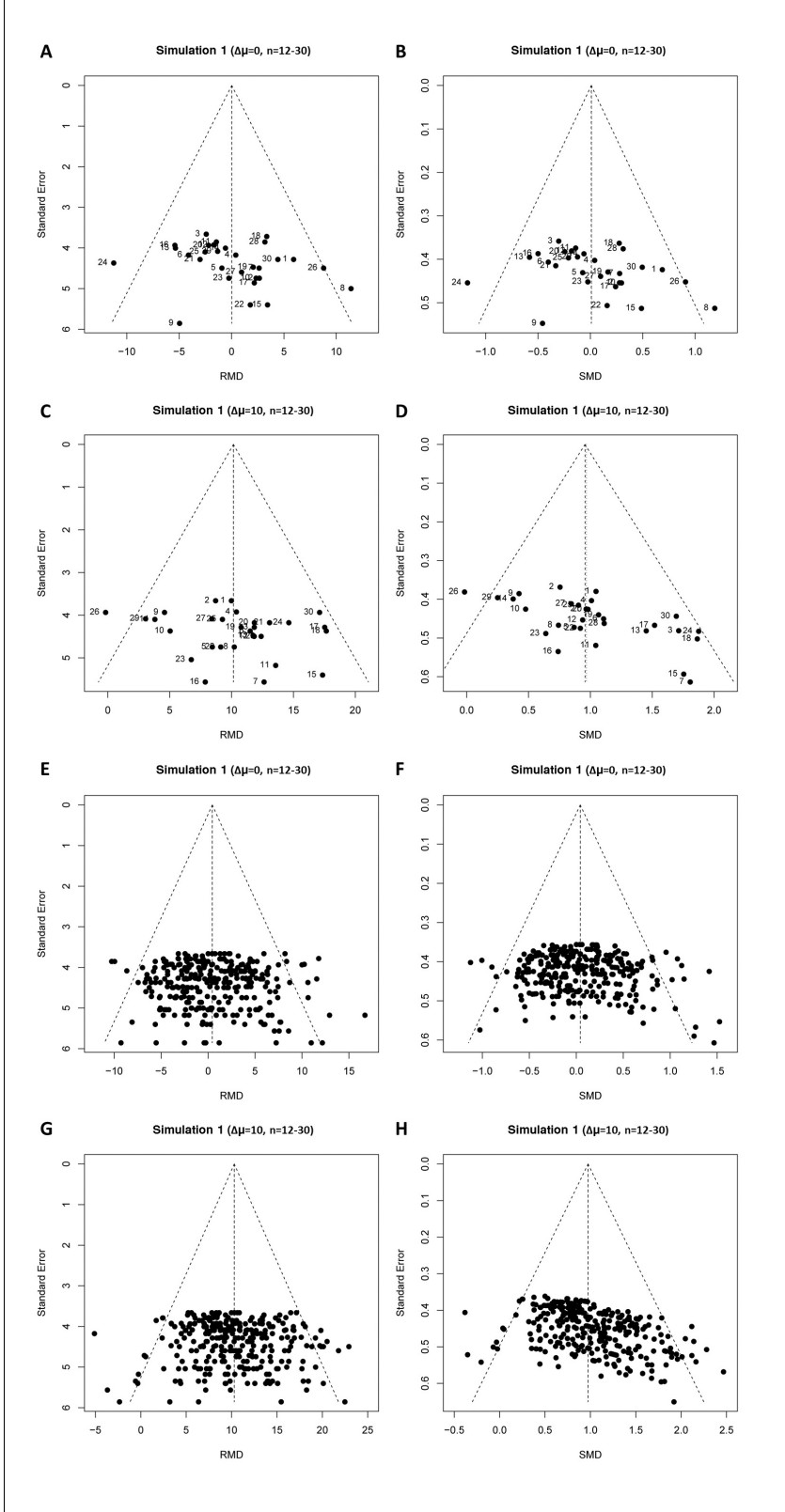

**Figure 4.** Representative raw mean difference (RMD; **A, C, E, G**) and standardized mean difference (Hedges' *g* SMD; **B, D, F, H**) funnel plots for simulated unbiased meta-analyses containing thirty (**A–D**) or 300 (**E–H**) studies with a small sample size (total study n = 12–30). Simulations were performed without an intervention effect (Δμ = 0; **A–B and E–F**), or with an intervention effect (Δμ = 10; **C–D and G–H**). Δμ = difference in normally distributed means between control and intervention group. Representative funnel plots for studies with a large sample size (total study n = 60–320) are

*Figure 4 continued on next page*

*Figure 4 continued*

shown in *Figure 4—figure supplement 1*. Representative funnel plots for the comparison between Hedges' *g* and Cohen's *d* are shown in *Figure 4—figure supplement 2*.

DOI: https://doi.org/10.7554/eLife.24260.005

The following figure supplements are available for figure 4:

**Figure supplement 1.** Simulation 1 funnel plots for large study sample sizes.

DOI: https://doi.org/10.7554/eLife.24260.006

**Figure supplement 2.** Funnel plots comparing Cohen's *d versus* Hedges *g*.

DOI: https://doi.org/10.7554/eLife.24260.007

true effect size. Finally, through visual inspection, distortion could be seen in all SMD funnel plots that incorporated a true effect, most prominent in the preclinical (small study) scenarios (*Figure 4* and *Figure 4—figure supplement 1*).

When repeating the simulations using Cohen's *d* SMD instead of Hedges' *g*, or using Begg and Mazumdar's test, we found highly similar results in all scenarios simulated (see *Supplementary file 1* and exemplary funnel plots in *Figure 4—figure supplement 2*).

Next, we assessed the impact of censoring non-significant simulated experiments (to simulate publication bias) and the performance of SMD vs. $1/\sqrt{n}$ funnel plots and NMD funnel plots in the presence of an intervention effect as alternatives to the SMD vs. SE funnel plot. As in simulation 1, SMD vs. SE funnel plots of unbiased simulations were identified as asymmetrical by Egger's test (*Table 2*). However, when the precision estimate was changed from SE to $1/\sqrt{n}$, the prevalence of false positive results fell to the expected 5% (*Table 2*). For the NMD, Egger's test performed correctly when using either the SE or $1/\sqrt{n}$ as precision estimate. In all scenario's, approximately 50 out of 1000 simulated funnel plots appeared to be asymmetrical by chance (*Table 2*). The results of Egger's test are supported by visual inspection of funnel plots of these unbiased scenario's (*Figure 5*). The typical left-upward shift of the small SMD datapoints and right-downward shift of the large SMD data points is clearly visible in the SMD vs. SE plot (*Figure 5B*), but not in the RMD, SMD vs. $1/\sqrt{n}$ or NMD plots.

In our final simulation we tested the performance of these different approaches in the presence of simulated publication bias. In the majority of these simulations of meta-analyses of individually small studies, asymmetry was detected both visually (*Figure 6*), and using Egger's regression (*Supplementary file 1*). When the size of individual studies was small, SMD vs.$1/\sqrt{n}$ funnel plots performed as well as the RMD vs. SE funnel plots, in both biased and unbiased simulations (*Table 2*). The NMD also behaved similar to the RMD with either an SE or $1/\sqrt{n}$ precision estimate.

**Table 2.** publication bias assessments in unbiased and biased simulations using the RMD, SMD or NMD in combination with an SE or sample size-based precision estimate (simulation 3).

| Effect measure | Bias? | Precision estimate SE | | Precision estimate $1/\sqrt{n}$ | |
|---|---|---|---|---|---|
| | | % of sims with Egger's p<0.05 | Median p-value (range) | % of sims with Egger's p<0.05 | Median p-value (range) |
| RMD | No | 5.1 | 0.51 (0.001–1.0) | 5.1% | 0.50 (0.001–1.0) |
| RMD | Yes | 69.1% | 0.01 ($2.7*10^{-8}$ - 0.99) | 69.6% | 0.01 ($1.6*10^{-8}$ - 0.97) |
| SMD | No | 100% | $2.9*10^{-13}$($0–8.1*10^{-6}$) | 4.3% | 0.51 (0.001–1.0) |
| SMD | Yes | 100% | $4.4*10^{-16}$($0–1.8*10^{-6}$) | 72.4% | 0.01 ($5.4*10^{-10}$ - 0.99) |
| NMD | No | 6.4% | 0.51 (0.001–1.0) | 6.4% | 0.50 (0.001–1.0) |
| NMD | Yes | 60.5% | 0.02 ($7.1*10^{-8}$ - 0.99) | 60.4% | 0.02 ($8.0*10^{-8}$ - 0.98) |

Simulated meta-analyses contained 300 studies (total study n = 12–30 subjects) and the difference in normally distributed means between control and intervention group was 10. Publication bias was introduced stepwise, by removing 10% of primary studies in which the difference between the intervention and control group means was significant at p<0.05, 50% of studies where the significance level was p≥0.05 to p<0.10, and 90% of studies where the significance level was p≥0.10. SE = standard error; RMD = raw mean difference; SMD = standardized mean difference (Hedges' g); NMD = normalized mean difference; sims = simulations.

DOI: https://doi.org/10.7554/eLife.24260.009

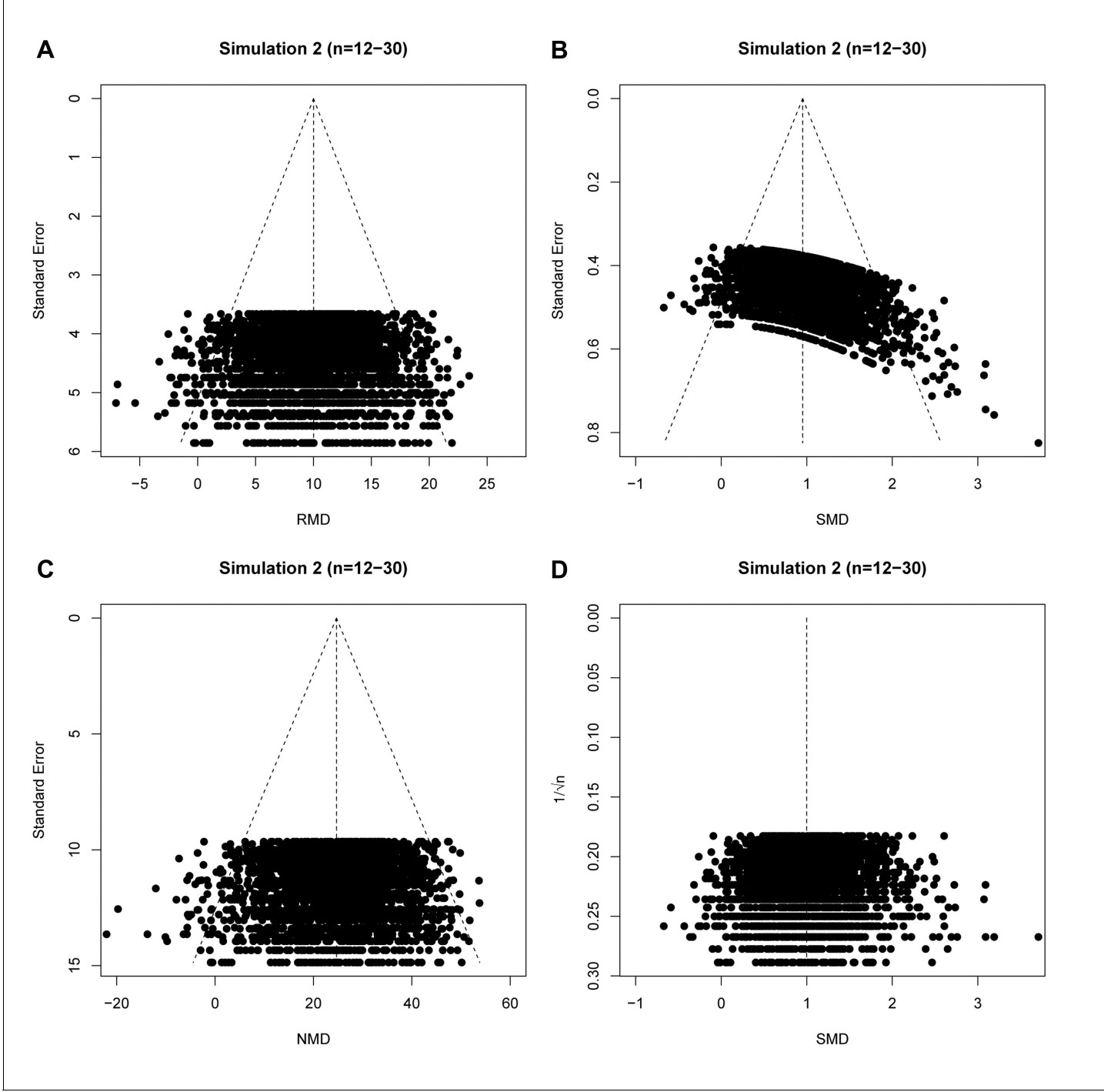

**Figure 5.** raw mean difference (RMD; **A**), standardized mean difference (SMD; **B**), normalized mean difference (NMD; **C**) with SE as precision estimate, and SMD funnel plots using $1/\sqrt{n}$ as precision estimate (**D**). All plots show the same simulated meta-analysis containing 3000 studies with small sample sizes (n = 12–30) and an overall intervention effect of $\Delta\mu = 10$. $\Delta\mu$ = difference in normally distributed means between control and intervention group.
DOI: https://doi.org/10.7554/eLife.24260.010

## Re-analyses of SMD funnel plots from published meta-analyses

Since a sample size-based precision estimate might be more suitable for asymmetry analysis, we used data from five previously published meta-analyses which used an SMD vs. SE funnel plot and claimed funnel plot asymmetry as a result of publication bias. In the original publications, all five of

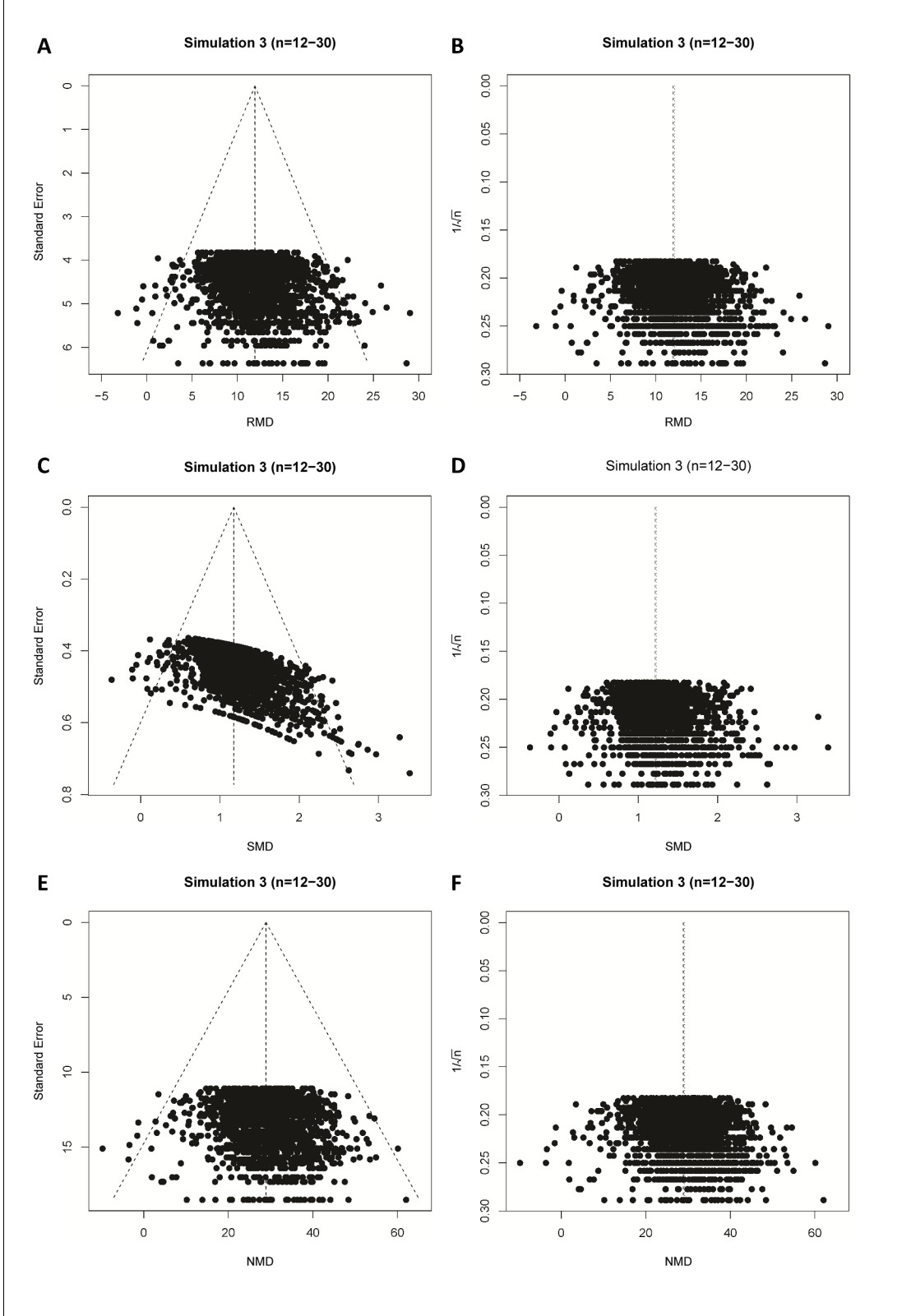

**Figure 6.** Simulation 3 funnel plots of biased meta-analyses. Representative funnel plots of simulated biased meta-analyses using a raw mean difference (RMD; **A–B**), a standardized mean difference (SMD; **C–D**), or a normalised mean difference (NMD; **E–F**) effect measure. The present example contains 3000 studies with a small study sample size (n = 12–30) and an intervention effect present (difference in normal distribution means between control and intervention group = 10). Publication bias was introduced stepwise, by removing 10% of primary studies in which the difference between

*Figure 6 continued on next page*

*Figure 6 continued*

the intervention and control group means was significant at p<0.05, 50% of studies where the significance level was p≥0.05 to p<0.10, and 90% of studies where the significance level was p≥0.10. Precision estimates are standard error (**A, C, E**) or sample size-based (**B, D, F**), where n = total primary study sample size.

DOI: https://doi.org/10.7554/eLife.24260.011

these funnel plots were asymmetrical according to Egger's regression test. In three out of five cases, this asymmetry was not present in funnel plots using $1/\sqrt{n}$ as a precision estimate (**Table 3** and **Figure 7**). Furthermore, three out of five papers reported several missing data points, as detected by trim and fill analysis. Missing data points were not detected when using SMD vs. $1/\sqrt{n}$ funnel plots for trim and fill analysis (**Table 3** and **Figure 7**).

## Discussion

Using data from both simulated and empirical meta-analyses, we have shown that the use of Egger's regression test for funnel plot asymmetry based on plotting SMD against SE is associated with such a substantial over-estimation of asymmetry as to render this approach of little value, particularly when the size of contributing studies is small. This distortion occurs whenever an intervention effect is present, in meta-analyses both with and without publication bias. The severity of distortion and the risk of misinterpretation are influenced by the sample size of the individual studies, the number of studies in the meta-analysis, and the presence or absence of an intervention effect. Thus, the use of SMD vs. SE funnel plots may lead to invalid conclusions about the presence or absence of publication bias and should not be used. Since it is the association between the SMD and its SE that leads to funnel plot distortion, it almost inevitable that the issues described will occur with any test for publication bias that relies on an assessment of funnel plot asymmetry (*e.g.* Begg and Mazumdar's test [**Begg and Mazumdar, 1994**]). When using trim and fill analysis, funnel plot distortion introduces the risk of incorrectly adjusting the summary effect estimate. Previous reports of the presence of publication bias based on this approach should be re-evaluated, both for pre-clinical and clinical meta-analyses. Importantly, distortion does not occur in NMD vs. SE funnel plots, which formed the basis of a recent analysis showing evidence for substantial publication bias in the animal stroke literature (**Sena et al., 2010**).

As the use of meta-analysis to summarize clinical and preclinical data continues to increase, continuous evaluation and development of research methods is crucial to promote high-quality meta-research (**Ioannidis et al., 2015**). To our knowledge (see also **Sterne et al., 2011**), potential problems in tests for funnel plot asymmetry have not been extensively studied for SMDs, and guidance is limited. For instance, the Cochrane Handbook for Systematic Reviews of Interventions (**Yan et al., 2015**) states that artefacts may occur and that firm guidance on this matter is not yet available. It is disquieting that publication bias analyses using SMD funnel plots have been published in clinical and preclinical research areas, presumably because both the authors and the peer reviewers were

**Table 3.** Re-analysis of published preclinical meta-analyses using SMD

| | | | Precision estimate | | | | | |
| | | | Standard Error | | | $1/\sqrt{n}$ | | |
| Study | n | Observed SMD[95% CI] | Egger's p | filled | Adjusted SMD | Egger's p | filled | Adjusted SMD |
|---|---|---|---|---|---|---|---|---|
| *Egan et al. (2016)* | 1392 | 0.75 [0.70, 0.80] | <2.2×10⁻¹⁶ | 252 | 0.42 [0.37,0.47] | $2.2 \times 10^{-11}$ | 0 | N/A |
| *Groenink et al. (2015)* | 43 | −1.99[−2.33,−1.64] | $8.5 \times 10^{-10}$ | 0 | N/A | 0.68 | 0 | N/A |
| *Kleikers et al. (2015)* | 20 | −1.15[−1.67; −0.63] | $3.5 \times 10^{-4}$ | 6 | ? | $2.9 \times 10^{-3}$ | 0 | N/A |
| *Wever et al. (2012)* | 62 | 1.54 [1.16, 1.93] | $7.8 \times 10^{-6}$ | 3 | ? | 0.62 | 0 | N/A |
| *Yan et al. (2015)* | 60 | 1.58 [1.19, 1.97] | $6.5 \times 10^{-6}$ | 0 | N/A | 0.19 | 0 | N/A |

n = number of studies; SMD = standardized mean difference; CI = confidence interval; Egger's p=p value for Egger's regression; adjusted SMD = SMD after trim and fill analysis; N/A = not applicable.

DOI: https://doi.org/10.7554/eLife.24260.012

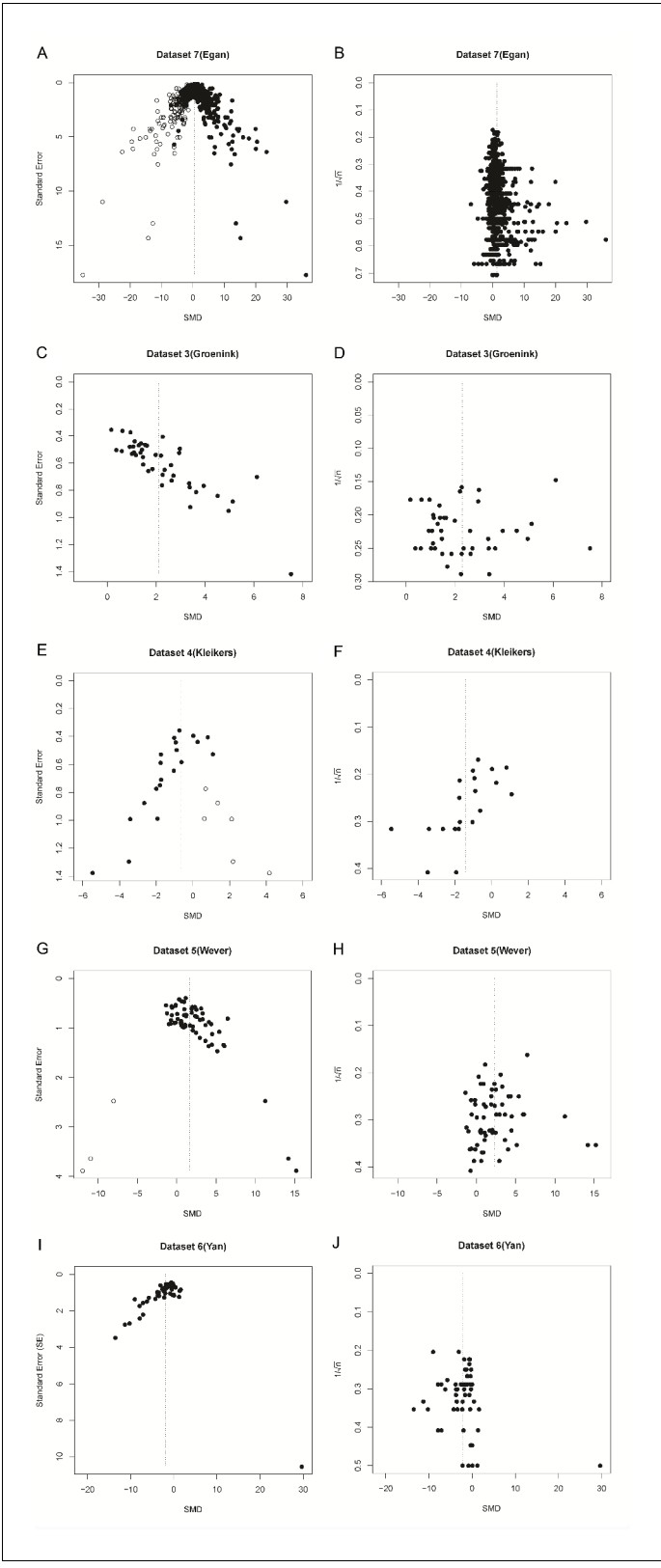

**Figure 7.** Funnel plots of re-analysis of empirical meta-analyses. Funnel plots of empirical meta-analyses plotted as standardized mean difference (SMD) *versus* standard error, as in the original publications (left hand panels), and as SMD *versus* $1/\sqrt{n}$ after re-analysis. n = total primary study sample size; filled circles = observed data points; open circles = missing data points as suggested by trim and fill analysis.
DOI: https://doi.org/10.7554/eLife.24260.013

unaware of the risk of spurious publication bias introduced by this methodology. Accepted papers from our group and others using SMDs for publication bias assessments have passed the peer review system, with no additional questions and or comments on this potential problem.

A similar phenomenon has been reported for the use of odds ratios in funnel plots, which also induces artificial significant results in Egger's regression (*Peters et al., 2006*). Here, too, an alternative test based on sample size has been proposed to circumvent this problem (*Peters et al., 2006*), and we suggest to extend this recommendation to SMDs.

However, given the relative performance of the RMD, NMD and SMD approaches, it is reasonable to consider whether SMD should ever be used. The RMD approach is limited because there are many instances (for example across species) where, although the same units of measurement are used, a given change may have very different biological importance. The NMD approach is preferred, but – because it expresses the effects of an intervention as a proportion of lesion size – there may be circumstances where outcome in a non-lesioned animal is not reported or cannot be inferred, and here the NMD approach is not possible. Further, the relative performance of RMD, NMD and SMD approaches in identifying heterogeneity between groups of animal studies (partitioning of heterogeneity) or in meta-regression is not known.

Taken with the increased distortion seen when contributing studies are individually small, this means our findings may be especially relevant for preclinical meta-analyses. The SMD is frequently used in preclinical meta-analyses to overcome expected heterogeneity between data obtained from different animal species. Nevertheless, the SMD is also used in clinical meta-analyses and the degree of distortion cannot be readily predicted. In any case, distortion causes the threshold for determining publication bias to be artificially lowered when using SMDs and their SE, increasing the chance of false-positive results.

Of note, trim and fill analysis may not always be reliable when the number of studies in a meta-analysis is large; in half of the cases of our unbiased simulations with 300 and 3000 studies, many studies were deemed missing, even if no intervention effect was introduced. Still, the SMD simulations were always more susceptible to the addition of imputed studies if a true effect was introduced, and the effect size reduction was larger compared to RMD measurements.

## Limitations of this study

We designed our data simulations to closely resemble empirical data in terms of the range of sample sizes, effect sizes and numbers of studies in a meta-analyses. We acknowledge that our current range of simulation scenarios does not enable us to predict the impact of funnel plot distortion in every possible scenario, but we present those scenarios which most clearly illustrate the causes and consequences of funnel plot distortion. Furthermore, our simulations may still be improved by *e.g.* studying the effects of unequal variances between treatment groups, sampling data from a non-normal distribution, or introducing various degrees of heterogeneity into the simulation. However, research on how to optimally simulate these parameters is first needed, and was beyond the scope of this study. instead, we used re-analyses of empirical data to test our proposed solutions on a number of real-life meta-analyses which include all of the aforementioned aspects.

## Recommendations

We recommend that, where possible, investigators use RMD or NMD instead of SMD when seeking evidence of publication bias in meta-analyses. Where it is necessary to use SMD, assessment for publication bias should use a sample size-based precision estimate such as $1/\sqrt{n}$. In a given analysis it may be possible to calculate an NMD effect size for some but not all studies. In these circumstances there is a trade-off between the reduced number of included studies and an improved estimation of publication bias, and sensitivity analysis may be used to compare the meta-analysis outcome using the NMD versus the SMD. Of note, other methods to investigate publication bias in a dataset may be used in addition to funnel plots (*e.g.* fail-safe N, Excess Significance Test [*Ioannidis and Trikalinos, 2007*], or selection method / weight funcion model approaches [*Peters et al., 2006*]), but the performance of these approaches in the context of SMD, RMD and NMD estimates of effect size is not known.

In conclusion, funnel plots based on SMDs and their SE should be interpreted with caution, as the chosen precision estimate is crucial for detection of real funnel plot asymmetry.

## Materials and methods

We performed data simulations and re-analyses of empirical data using R statistical software (version 3.1.2; RRID:SCR_001905) and the most recent MBESS, xlsx, meta and metafor packages (*Rothstein et al., 2005*; *Kelley, 2016*; *Schwarzer, 2016*; *Viechtbauer, 2010*; *Dragulescu, 2014*) (See *Supplementary file 3* for all R scripts). For all analyses involving RMD and SMD the primary outcome of interest was the number of asymmetrical funnel plots as detected by Egger's regression (*Egger et al., 1997*). As a secondary outcome, we assessed the number of missing studies as imputed by Duval and Tweedie's trim and fill analysis (*Duval and Tweedie, 2000*). This method provides an estimate of the number of missing studies in a meta-analysis, and the effect that these missing studies may have had on its outcome. In brief, the funnel plot is mirrored around the axis represented by the overall effect estimate. Excess studies (often small, imprecise studies with a neutral or negative effect size) which have no counterpart on the opposite side of the plot are temporarily removed (trimmed). The trimmed plot is then used to re-estimate the overall effect estimate. The trimmed data points are placed back into the plot, and then a paired study is imputed with the same precision but reflected to have an effect size reflected around the adjusted overall estimate, and plotted in a different color or symbol from the observed data points. The analysis is re-run and repeated until no further asymmetry is observed. We used trim and fill analysis and a random effects model in R to seek evidence for publication bias overstating the effectiveness of the interventions, based on the proposed direction of the intervention effect. Because of its superior performance in studies with small sample sizes, Hedges' *g* was used in the main analyses throughout this manuscript. We considered a p-value of <0.05 to be significant for Egger's regression in individual simulations.

### Empirical data published as RMD re-analyzed as SMD

In our first re-analysis of empirical data from published preclinical meta-analyses (*Wever et al., 2015*; *Zwetsloot et al., 2016*), we constructed funnel plots using the unbiased SMD (Hedges' *g* [*Hedges, 1981*]) vs. SE, and compared these to funnel plots using the RMD vs. SE (as in the original publication).

### Data simulation methods

In our first simulation, we tested the estimation of publication bias using the unbiased SMD (Hedges' *g*) in simulated data where there was no publication bias. As a sensitivity analysis, all scenarios of simulation 1 were also performed using Cohen's *d*. We generated simulated meta-analyses by simulating the desired number of individual studies, each with a control group and an intervention group. The control groups were simulated by randomly sampling individual subject data from a normal distribution with a mean ($M_{ctrl}$) of 30 and an SD of 10 (*Table 4*); these values were based on outcome data for functional imaging in myocardial infarction studies (*Zwetsloot et al., 2016*). Individual subject data for the intervention group was sampled from a normal distribution with mean $M_{ctr}$ +ES (effect size). To assess the effect of differences in overall intervention effects on funnel plot distortion, we simulated meta-analyses for an ES of respectively 0, 5, or 10 (*Table 4*). To assess the effect of study sample size on funnel plot distortion, we simulated two types of study sizes: small (12–30 subjects per study), as is more common in animal studies, and large (60–320 subjects per study), as

**Table 4.** Simulation characteristics.

| Experimental groups | Small studies | | | Large studies | | | RMD | SMD | NMD |
|---|---|---|---|---|---|---|---|---|---|
| | *N* | *Mean* | *SD* | *N* | *Mean* | *SD* | | | |
| Intervention 1 (no effect) | 7–14 | 30 | 10 | 40–150 | 30 | 10 | 0 | 0 | 0 |
| Intervention 2 (RMD = 5) | 7–14 | 35 | 10 | 40–150 | 35 | 10 | 5 | 0.5 | 0.125 |
| Intervention 3 (RMD = 10) | 7–14 | 40 | 10 | 40–150 | 40 | 10 | 10 | 1 | 0.25 |
| Control | 5–16* | 30 | 10 | 20–170* | 30 | 10 | | | |
| Sham | 4–6 | 70 | 4 | | | | | | |

n = sample size; ND = normal distribution; SD = standard deviation; *control group sample size = intervention group sample size $\pm \leq 2$ (small studies) or $\pm \leq 20$ (large studies).

DOI: https://doi.org/10.7554/eLife.24260.014

is more common in human studies. For each simulated study, we determined the number of subjects by sampling the group sizes from the uniform distribution within the ranges of study sizes given (*Table 4*). Of note, an intervention effect of SMD = 1 may appear large to those experienced in meta-analyses of clinical data, but is typical of those observed in animal studies, as are the group sizes reported (see e.g. *Figure 2* and *Table 3*).

Simulation and aggregation of individual subject data into study-level data was repeated until the desired number of studies to be included in the meta-analysis was obtained. We assessed the influence of the number of included studies on funnel plot distortion by simulating meta-analyses containing either 30, 300, or 3000 studies. Although there is no consensus on the minimal number of studies required for publication bias analysis, 30 has been previously proposed as the minimal number to obtain sufficient power for asymmetry testing (*Lau et al., 2006*). We chose 3000 studies for the largest meta-analysis as this is substantially larger than any meta-analysis of which we know, and any effects of study number are likely to be saturated at that number of studies. Importantly, we did not introduce publication bias to any of these datasets and the funnel plots should therefore be symmetrical. We repeated each simulation 1000 times, and we compared the effects of expressing the meta-analysis results as RMD or SMD, and used funnel plots with the effects size plotted on the x-axis and the SE as precision estimate plotted on the y-axis (RMD vs. SE and SMD vs. SE plots). As a second sensitivity analysis, we assessed the robustness of our findings using Egger's test by re-testing all scenario's of simulation 1 using Begg and Mazumdar's test (*Begg and Mazumdar, 1994*).

Informed by the outcomes of simulation 1, in our second simulation we selected the conditions introducing the most prominent distortion in SMD vs. SE funnel plots to investigate the performance of alternatives including SMD vs. $1/\sqrt{n}$ funnel plots and NMD funnel plots. Thus, all simulations were performed with a small study sample size, in the presence of an intervention effect (see *Table 4*) and with 3000 studies per meta-analysis. Under these conditions, we constructed RMD vs. SE and SMD vs. SE funnel plots as described above, as well as funnel plots of the SMD against the inversed square root of the total sample size ($1/\sqrt{n}$) in each study, and of the NMD against the SE. For the NMD, sham group data were simulated to have a mean of 70 and an SD of 4 (*Table 4*). Group size was selected to be 4–6 subjects, which is a typical sample size for sham groups in preclinical experiments. We performed the simulations once and compared outcomes across all four funnel plots.

In our final simulation we investigated the effects of a modelled publication bias on the performance of the SMD vs. SE and alternative approaches. We simulated meta-analyses containing 300 and 3000 studies with a small individual sample size and an intervention effect present ($\Delta\mu$ = difference in means between control and intervention group = 10; see *Table 4*). RMD vs. SE, RMD vs. $1/\sqrt{n}$, SMD vs. SE, SMD vs. $1/\sqrt{n}$ and NMD vs. SE funnel plots were constructed and tested for asymmetry using Egger's regression. We then introduced publication bias in these meta-analyses using a stepwise method, Publication bias was introduced stepwise, by removing 10% of primary studies in which the difference between the intervention and control group means was significant at p<0.05 (Student-t test), 50% of studies where the significance level was p≥0.05 to p<0.10, and 90% of studies where the significance level was p≥0.10. Funnel plot asymmetry testing was performed as above, and the results were compared to the unbiased simulations and between different funnel plot types. All simulations were repeated 1000 times. Of note, this simulation was not performed for meta-analyses of studies with a large sample size, since pilot data showed that the large sample size will cause only very few studies to be removed from the 'biased' meta-analysis.

## Re-analysis of empirical data using an n-based precision estimate

Finally, to assess the usefulness and impact of using a sample size-based precision estimate in SMD funnel plots of empirical data, we re-analysed data from five published preclinical meta-analyses that used SMD vs. SE funnel plots to assess publication bias. The selected datasets were from our own groups, or from recent collaborations, which allowed for easy identification of meta-analyses using SMD vs. SE funnel plots, and easy access to the data. There were no selection criteria in terms of *e. g.* the number of studies in the analysis, or the outcome of the publication bias assessment. The distribution of the total number of subjects per data point in the selected studies is (in median (min-max): 11.7 (6–38) for *Wever et al. (2012)*, 20(12-46) for *Groenink et al. (2015)*, 11(4-24) for *Yan et al., 2015*, 14.5 (6–35) for *Kleikers et al. (2015)* and 12(4-66) for *Egan et al. (2016)*. For these data sets, we compared the outcome of Egger's regression and trim and fill analysis when using

SMD vs. SE funnel plots to that of SMD vs. $1/\sqrt{n}$ funnel plots. We obtained the corresponding author's consent for re-analysis.

## Additional information

### Funding

| Funder | Grant reference number | Author |
|---|---|---|
| National Institute of Environmental Health Sciences | National Toxicology Program research funding | Kimberley E Wever |
| Netherlands Cardiovascular Research Initiative | CVON-HUSTCARE | Steven AJ Chamuleau |
| National Centre for the Replacement, Refinement and Reduction of Animals in Research | Infrastructure Award | Emily S Sena<br>Malcolm R MacLeod |
| Alexander Suerman Program | PhD student Scholarship | Peter-Paul Zwetsloot |

The funders had no role in study design, data collection and interpretation, or the decision to submit the work for publication.

### Author contributions

Peter-Paul Zwetsloot, Mira Van Der Naald, Conceptualization, Data curation, Formal analysis, Investigation, Methodology, Writing—original draft, Writing—review and editing; Emily S Sena, David W Howells, Conceptualization, Writing—review and editing; Joanna IntHout, Methodology, Writing—review and editing; Joris AH De Groot, Formal analysis, Methodology, Writing—review and editing; Steven AJ Chamuleau, Supervision, Funding acquisition, Writing—review and editing; Malcolm R MacLeod, Conceptualization, Supervision, Writing—review and editing; Kimberley E Wever, Conceptualization, Data curation, Formal analysis, Supervision, Funding acquisition, Methodology, Writing—original draft, Project administration, Writing—review and editing

### Author ORCIDs

Emily S Sena iD http://orcid.org/0000-0002-3282-8502
Malcolm R MacLeod iD http://orcid.org/0000-0001-9187-9839
Kimberley E Wever iD http://orcid.org/0000-0003-3635-3660

### Decision letter and Author response

Decision letter https://doi.org/10.7554/eLife.24260.019
Author response https://doi.org/10.7554/eLife.24260.020

## Additional files

### Supplementary files

• Supplementary file 1. Cohen's *d* and Hedges' *g*, as well as Egger's test and Begg and Mazumdar's test, perform similar in multiple illustrative scenario's (simulation 1).
DOI: https://doi.org/10.7554/eLife.24260.015

• Supplementary file 2. Supplemental equations.
DOI: https://doi.org/10.7554/eLife.24260.016

• Supplementary file 3. R scripts.
DOI: https://doi.org/10.7554/eLife.24260.017

• Transparent reporting form
DOI: https://doi.org/10.7554/eLife.24260.018

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
