## [Decision Letter]

[Editors’ note: this article was originally rejected after discussions between the reviewers, but the authors were invited to resubmit after an appeal against the decision.]

Thank you for submitting your work entitled "Standardized mean differences cause funnel plot distortion in publication bias assessments" for consideration by *eLife*. Your article has been favorably evaluated by a Senior Editor and three reviewers, one of whom, M Dawn Teare (Reviewer #1), is a member of our Board of Reviewing Editors. The following individuals involved in review of your submission have agreed to reveal their identity: Marcus Munafo (Reviewer #2); Jack Vevea (Reviewer #3).

The reviewers have recognised this is an important topic but you have not really presented what the impact of this problem is. There are many queries relating to selection of simulation parameters, choice of statistic and test, and what the impact is in realistic practice.

Addressing these issues will likely exceed the time span for major revisions in *eLife* (two-three months), so we are rejecting the paper now but would welcome a resubmission in the future, with no guarantees of acceptance or re-review. This should be a de novo submission, and we would endeavour to recruit the same editors to assess the revisions.

*Reviewer #1:*

This manuscript identifies an important potential problem when using the SMD in meta-analyses. When sample sizes are small and effect sizes large, funnel plots using SMD vs. SE can show asymmetry and hence suggest evidence of publication bias when there is none.

The authors demonstrate that this issue should not be so surprising as the SMD SE is a function of the effect size and hence studies with small sample sizes will tend to show a bias. Meta analyses that contain large numbers of studies are therefore more likely to show this effect. This could be an important issue as the meta-analysis may over correct the SMD and actually lead to an underestimate of an effect size.

This manuscript has a clear simple message that using funnel plots of the SMD vs. 1/√n do not result in the same bias.

I do have a number of concerns with the paper especially as they do not tackle the issue of what is the impact on the overall estimate of the effect size resulting from such meta-analyses. My understanding is that the funnel plot, trim and fill is used to estimate corrected effect sizes (so if publication bias is suspected the SMD is adjusted). This manuscript has not really focused on the impact on the overall estimates coming out of meta-analyses.

1) Hedges' *g* is often reported for studies of small sample size rather than Cohen's *d*, but the authors appear to have not performed their simulations using this version. (Apart from a single figure shown in the supplementary methods). It seems odd not to have fully evaluated the performance of both summary measures.

2) The simulations have only been performed under quite a limited number of scenarios, the null, for one very large effect size and then for a specific form of publication bias. Though a range of sample sizes has been explored. While this is helpful to show the weakness in the method researchers will want to know when it is important. Is an effect size of 1SD a realistic effect size in practice? Surely if an effect size is so large as that not many studies will be required to confirm it. Table 3 shows scant details of the 5 meta-analyses. It seems intriguing that one of these meta-analyses included almost 1400 studies? I have looked up that reference and while they have looked for evidence of publication bias in all the various studies, it does not seem sensible to have pooled all of the studies for this analysis.

3) Much more detail on the meta analyses and why they were selected. What was the impact on the estimates of SMD using the different funnel plots? What was the distributions of sample sizes and SMDs in the studies making up each meta-analysis?

*Reviewer #2:*

The authors highlight an interesting aspect of commonly used tests to assess for the presence of small study bias, which may be caused by publication bias. This is supported by analyses of simulated and real data.

My main comment is that there are other tests (e.g. Begg and Mazumdar). Do the issues described here apply to all tests, and if so does the extent to which they do so differ (in which case, which is superior)?

It may also be worth briefly discussing other approaches, such as the Excess Significance Test developed by Ioannidis, that rely on different assumptions and therefore allow triangulation of methods.

*Reviewer #3:*

This manuscript presents a simulation study looking at the identification of publication bias for measures of effect size based on the difference between means: standardized mean difference (SMD), raw mean difference (RMD), and normalized mean difference (NMD). Using Egger's regression and Duval and Tweedie's trim and fill analysis to detect funnel plot asymmetry, the authors discover a problem of overidentification for SMD when analysis is based on funnel plots of SMD against standard error. Overidentification of publication bias was found to be much greater for SMD in comparison to RMD and NMD. This problem was found to be mitigated by plotting SMD against 1/√n.

The authors provide a good description of the different types of measures of effect size. However, they provide only a mention of Hedges' *g* in the fourth paragraph of Section 1.1. Many practitioners incorrectly use the term *g* and *d* interchangeably, and a description of how these two estimates are different will be beneficial. Related to this, in the following sentence the paper describes the shortcomings of SMD when the sample size is small. It would be useful to point out that Hedges' bias-corrected estimate is meant to address bias when sample sizes are small (Hedges, 1981).

In Section 1.2, the authors describe the relationship between funnel plot asymmetry and publication bias. It would also be worth pointing out that funnel plot asymmetry does not necessarily indicate publication bias. There are other reasons why funnel plots may appear asymmetrical, such as systematic heterogeneity related to the inclusion of two different modes of inquiry that differ both in effect magnitude and in typical standard error.

There are two sections of the paper with the heading "Data Simulations" (Section 2.2 and Section 4.1). Results of the simulations are in Section 2.2. Section 4.1 is at the end of the paper and describes the process for simulating the data. This organization is confusing. It would be better if the description of the simulation were provided before these results, as it provides context for understanding the outcome of the author's simulation. Also, headings that clearly define these different sections will make it easier for readers to navigate the article.

In the third paragraph of Section 2.2 the authors state that results were similar for Cohen's *d* and Hedges' *g* effect-size estimates. Based on the information in the paper (Figure 3 legend and Figure 3—figure supplement 2), it appears that the comparisons for these estimates were only made using large sample sizes. In this case, similar results would be expected. They would be more likely to have different results in cases where sample sizes are small, as *d* and *g* are more-or-less identical for large sample sizes. The authors should address this scenario as well and make it clear that the information provided uses the Hedges’ *g* bias-corrected estimate of effect size.

In the fourth paragraph of Section 2.2 the authors use SMD-1/√n to refer to their analysis of funnel plots looking at SMD with 1/√n on the y axis. The hyphen could be interpreted as arithmetic minus sign, so additional clarity of notation is needed.

In the fourth paragraph of Section 2.2 the authors state that the distortion is not seen in plot D of Figure 4. Visual interpretation of funnel plots is subjective. To some readers, the plot will continue to appear asymmetric, but flatter. The authors should explain what details of the plot lead them to the conclusion that the plot was symmetric as well as plots A and C, which are also described as not indicating distortion.

In Table 2, the authors of this study operationalize publication bias by removing all studies with a p-value ≥.10. This operation means that studies with a p-value above the cutoff have no chance of being published, and all studies with a p-value below (e.g. p=.09 and p=.01) have equal certainty of publication. This specification is not a good reflection of how publication bias functions in the real world. The authors should provide justification for using a simple cutoff of a p-value, rather than a model with diminishing probability such as a step function or a decaying continuous function. The authors should also include their reasoning for setting the cutoff at p ≥}.10, as well as information on how many studies were excluded based on this cutoff, such as the average number of studies remaining in the analysis.

In Section 2.3, the authors compare the results of the analyses using examples from real-world studies. Providing this shows how their research may be applicable in real world settings. It would also be useful to report effect-size estimate from the original meta-analysis.

The Discussion section is missing a discussion of the limitations of the current study. The simulation had a limited number of conditions. Providing limitations identifies the scope of the presented findings. Also, the authors should include a rationale explaining why they got these results Without that, the findings atheoretical; entirely empirical methodological findings are harder to accept with confidence.

In the first paragraph of Section 4.1, the authors describe the process for obtaining individual study sample sizes in the simulation. They should provide a rationale for sampling study sample sizes from a uniform distribution. Sample sizes typically do not follow a uniform distribution; rather they tend to be positively skewed. This brings into question the relevance of the simulations.

In Table 4 legend the authors describe the magnitudes for the effect sizes included in the simulation. They should provide a reasoning for using an effect magnitude of SMD=1. By many standards, this would be considered a very large effect. It is possible that the method would perform better (or worse) under conditions where the population magnitude is small or moderate.

Also, the simulation did not include heterogeneity of studies in the model: all studies were sampled from exactly the same fixed-effect distribution. Studies conducted in the real world tend to have heterogeneity. Conditions of heterogeneity often lead to failure of trim and fill and Eggers regression, and would influence the outcome of an analysis of publication bias.

---

## [Author Response]

[Editors’ note: the author responses to the first round of peer review follow.]

Reviewer #1:[…] I do have a number of concerns with the paper especially as they do not tackle the issue of what is the impact on the overall estimate of the effect size resulting from such meta-analyses. My understanding is that the funnel plot, trim and fill is used to estimate corrected effect sizes (so if publication bias is suspected the SMD is adjusted). This manuscript has not really focused on the impact on the overall estimates coming out of meta-analyses.

Although not our primary outcome, we have added the results of trim and fill analyses for simulation 1 to Table 1. These results indicate that using trim and fill in SMD vs. SE funnel plots will, in the majority of scenarios tested, lead to an overestimation of the number of missing studies and, as a result, and underestimation of the overall effect estimate after trim and fill. We have added these results to the manuscript text and Table 1.

Amendments to the manuscript:

Section 2.2: “Trim and fill analysis resulted in on average 7% extra studies filled in preclinical simulation scenarios using the RMD. […] As a result, the adjusted overall effect estimate after trim and fill in SMD funnel plots tended to be an underestimation of the true effect size.”

Added trim and fill results to Table 1

1) Hedges' g is often reported for studies of small sample size rather than Cohen's d, but the authors appear to have not performed their simulations using this version. (Apart from a single figure shown in the supplementary methods). It seems odd not to have fully evaluated the performance of both summary measures.

Indeed, Hedges’ *g* provides a more appropriate estimate of the SMD for studies with small sample sizes, which is why all simulations and re-analyses were performed in Hedges’ *g*. We infer from your question that this aspect of the methodology was not stated clearly enough, and have therefore improved the description in the first paragraph of the Materials and methods and Results sections to make it more clear that we used Hedges’ *g* in all main analyses.

We also agree that it is worthwhile to report the comparison between Hedges’ *g* and Cohen’s *d* more extensively. We have therefore extended Supplementary file 1 to include the results for all scenarios of simulation 1 in Hedges’ *g* and Cohen’s *d*, so that these can be easily compared. We have also extended Figure 4—figure supplement 2 with additional plots in both Hedges’ *g* and Cohen’s *d*. The data presented provide further support for our conclusion that the differences between the results of simulations using Hedges’ *g* versus those using Cohen’s *d* are marginal (see also Supplementary file 1).

Related to this, we have also added a more detailed description of the difference between Hedges’ *g* and Cohen’s *d*, because the two estimates are often confused (see reviewer 3, first comment).

Amendment(s) to the manuscript:

Added Supplementary file 1: Cohen’s *d* and Hedges’ *g*, as well as Egger’s test and Begg and Mazumdar’s test, perform similar in multiple illustrative scenario’s (simulation 1)

Section 2.2: “When repeating the simulations using Cohen’s d SMD instead of Hedges’ *g*, or using Begg and Mazumdar’s test, we found highly similar results in all scenarios simulated (see Supplementary file 1 and exemplary funnel plots in Figure 4—figure supplement 2).”

Section 4: “Because of its superior performance in studies with small sample sizes, Hedges’ *g* was used in the main analyses throughout this manuscript.”

Section 4.2: “[…] the unbiased SMD (Hedges’ *g*) […]. As a sensitivity analysis, all scenarios of simulation 1 were also performed using Cohen’s *d*.”

2) The simulations have only been performed under quite a limited number of scenarios, the null, for one very large effect size and then for a specific form of publication bias. Though a range of sample sizes has been explored. While this is helpful to show the weakness in the method researchers will want to know when it is important. Is an effect size of 1SD a realistic effect size in practice? Surely if an effect size is so large as that not many studies will be required to confirm it.

The chosen intervention effect size RMD = 10 / SMD =1 was based on empirical data from preclinical meta-analyses, where such large effects are commonly observed. For instance, all meta-analyses re-analyzed in this manuscript have overall effect sizes (considerably) larger than RMD=10 (Figure 1) or SMD=1 (Table 3). However, we appreciate that this effect size may seem very large to investigators accustomed to clinical meta-analyses, where effect sizes are often much smaller. We therefore appreciate your suggestion to extend our simulation scenarios with a smaller intervention effect size, and have performed additional simulations with an effect size of RMD=5 / SMD = 0.5. The results are presented in Table 1 (and also Supplementary file 1). We show that funnel plot distortion can also occur in these scenarios. The number of positive Egger’s tests in these unbiased simulations increased (sometimes dramatically) in nearly all scenario’s, regardless of the inserted effect size.

We have also added a sentence explaining the rationale behind the chosen effect sizes to Section 4.1. Furthermore, we have added the fact that our simulations do not cover the entire spectrum of possible simulations scenarios as a limitation of this study under Section 3.1 Limitations.

Amendment(s) to the manuscript:

Section 1: “Importantly, preclinical studies are, generally, individually small, with large numbers of studies included in meta-analysis, and large observed effects of interventions. This contrasts with clinical research, where meta-analyses usually involve a smaller number of individually larger experiments with smaller intervention effects.”

Section 2.2: updated text and Table 1 with RMD =5 / SMD = 0.5 simulation results.

Section 4.2: updated text and Table 4 with methodological details on new simulation scenarios with RMD =5.

Section 4.2: “Of note, an intervention effect of SMD = 1 may appear large to those experienced in meta-analyses of clinical data, but is typical of those observed in animal studies, as are the group sizes reported (see e.g. Figure 2 and Table 3).”

Section 3.1: “We acknowledge that our current range of simulation scenarios does not enable us to predict the impact of funnel plot distortion in every possible scenario, but we present those scenarios which most clearly illustrate the causes and consequences of funnel plot distortion.”

Table 3 shows scant details of the 5 meta-analyses. It seems intriguing that one of these meta-analyses included almost 1400 studies? I have looked up that reference and while they have looked for evidence of publication bias in all the various studies, it does not seem sensible to have pooled all of the studies for this analysis.

We agree with the reviewer that the decision whether or not to pool heterogeneous results from animal studies is a recurrent matter of discussion, and should be carefully considered for each meta-analysis of animal studies. However, our goal here is not to assess whether the authors of the published meta-analysis made the correct decision in pooling these 1400 studies for their publications bias assessment, but rather to show that their use of an SMD versus SE funnel plot led to an overestimation of publications bias in their analysis. We chose this example especially because the pattern of skewing in the published SMD versus SE plot led to a high number of imputed studies. This is why we present the authors’ published analysis as published, and compare the results with the SMD versus 1/√n analysis.

3) Much more detail on the meta analyses and why they were selected.

The selected datasets were from our own groups, or from our recent collaborations. This allowed for easy identification of meta-analyses using SMD vs. SE funnel plots, and easy access to the data required for re-analysis. We did not apply any selection criteria to the datasets, other than consent of the corresponding author. We limited the number of empirical examples to 5, because we felt that any more examples would be redundant. In addition to these practical advantages, we feel that we present the need for implementation of our approach best by reflecting critically on our own past work and point out limitations, rather than by directly criticizing the work of others. We have provided additional information in the text to better describe how the datasets were selected.

Amendment(s) to the manuscript:

Section 4.3: “The selected datasets were from our own groups, or from recent collaborations, which allowed for easy identification of meta-analyses using SMD vs. SE funnel plots, and easy access to the data. There were no selection criteria in terms of e.g. the number of studies in the analysis, or the outcome of the publication bias assessment.”

What was the impact on the estimates of SMD using the different funnel plots? What was the distributions of sample sizes and SMDs in the studies making up each meta-analysis?

We have now reported the original overall effect estimate, as well as the adjusted overall effect estimates after trim and fill in Table 3.

The range of SMDs in the studies making up each meta-analysis can be most easily examined using the funnel plots in Figure 6, by looking at their distributions along the x-axis. The distribution of the total number of subjects per data point in the selected studies is (in median(min-max)): 11.7(6-38) for Wever et al.2012, 20(12-46) for Groenink et al.2014, 11(4-24) for Yan et al. 2015, 14.5(6-35) for Kleikers et al. 2015 and 12(4-66) for Egan et al. 2016. We have added these data to the Materials and methods section. These ranges of sample sizes are also reflected in the y-axis of those funnel plots where 1/√n is used a precision estimate.

Amendment(s) to the manuscript:

Section 4.3: “The distribution of the total number of subjects per datapoint in the selected studies is (in median(min-max)): 11.7(6-38) for Wever et al.2012, 20(12-46) for Groenink et al.2014, 11(424) for Yan et al. 2015, 14.5(6-35) for Kleikers et al. 2015 and 12(4-66) for Egan et al. 2016”

Reviewer #2:[…] My main comment is that there are other tests (e.g. Begg and Mazumdar). Do the issues described here apply to all tests, and if so does the extent to which they do so differ (in which case, which is superior)?

Thank you for this interesting suggestion. We focus on the performance of Egger’s test since this is the test most commonly used in practice. However, based on our theoretical explanation of how the association between the SMD and its SE leads to funnel plot distortion, it is almost inevitable that the issues described will occur with any test that relies on an assessment of funnel plot asymmetry to detect publication bias (such as the method of Begg and Mazumdar, and others). As an example, we have investigated the performance of Begg and Mazumdar’s test, the results of which are shown in Supplementary file 1. We found that this test indeed results in overestimation of funnel plot asymmetry, highly similar to Egger’s regression. It seems to yield slightly lower number of false-positive results in scenarios with few studies, but this is likely due to the fact that Begg and Mazumdar’s test has been shown to have lower power than Egger’s test when the number of studies is low (Sterne et al. 2000 JCE). We propose to present the results of Begg and Mazumdar’s test for simulation 1 in Supplementary file 1, and to mention the result of this analysis in brief in the results and Discussion sections. We have made additions to the text accordingly.

For the reviewers interest, we present below the results for Table 2 using Begg and Mazumdar’s test, which also shows only marginal differences with those of Egger’s test (Table 2).

Effect measure**Bias?**
Precision estimate SEPrecision estimate 1/√n% of sims with Begg’s p<0.05median p-value (range)% of sims with Begg’s p<0.05median p-value (range)RMDNo5.40.52 (0.001 – 1.0)5.1%0.51 (0.002 – 1.0)RMDYes75.5%0.01 (3.6*10^-10^ – 0.89)76.3%0.01 (1.7*10^-10^ – 0.99)SMDNo100%6.2*10^-13^ (0 – 1.4*10^-4^)5.4%0.51 (0.002 – 1.0)SMDYes100%1.3*10^-14^ (0 – 6.5*10^-8^)82.4%0.003 (2.9*10^-10^ – 0.67)NMDNo6.2%0.51 (0.001 – 1.0)6.1%0.51 (0.001 – 1.0)NMDYes68.0%0.01 (8.1*10^-9^ – 0.96)68.6%0.01 (6.8*10^-9^ – 0.99)

Amendment(s) to the manuscript:

Supplementary file 1: results of Begg and Mazumdar’s test are now included for simulation 1.

Section 4.2: “As a second sensitivity analysis, we assessed the robustness of our findings using Egger’s test by re-testing all scenarios of simulation 1 using Begg and Mazumdar’s test (Wu et al., 2014), and comparing the results.”

Section 2.2:” When repeating the simulations using Cohen’s *d* SMD instead of Hedges’ *g*, or using Begg and Mazumdar’s test, we found highly similar results in all scenarios simulated (see Supplementary file 1 and exemplary funnel plots in Figure 4—figure supplement 2).”

Section 3: “Since it is the association between the SMD and its SE that leads to funnel plot distortion, it almost inevitable that the issues described will occur with any test for publication bias that relies on an assessment of funnel plot asymmetry (e.g. Begg and Mazumdar’s test (Yan et al., 2015)).”

It may also be worth briefly discussing other approaches, such as the Excess Significance Test developed by Ioannidis, that rely on different assumptions and therefore allow triangulation of methods.

The aim of our study is to promote the correct use of funnel plots for publication bias assessments. We agree that methods not relying on funnel plots may provide suitable alternatives, and have discussed such methods in Section 3.2 Recommendations. We agree that the Excess Significance Test should be mentioned here as well and have added it to the list of proposed alternatives. Of note, the performance of this test with continuous outcome measures has not been thoroughly evaluated, which is why we recommend it to be used cautiously.

Amendment(s) to the manuscript:

Section 3.2: “[…] Excess Significance Test […]”

Reviewer #3:[…] The authors provide a good description of the different types of measures of effect size. However, they provide only a mention of Hedges' g in the fourth paragraph of Section 1.1. Many practitioners incorrectly use the term g and d interchangeably, and a description of how these two estimates are different will be beneficial.

Thank you for your compliment and suggestion, we have added a more detailed description of the difference between Hedges’ *g* and Cohen’s *d* to the Introduction.

Amendment(s) to the manuscript:

Section 1.1:“Of note, equations 3 and 5 estimate the SMD using the approach of Cohen (Cohen, 1988); this estimate is therefore termed Cohen’s *d*. […] In many clinical meta-analyses, Hedges’ *g* will be almost identical to Cohen’s *d*, but the difference between the estimates can be larger in preclinical meta-analyses, where small sample sizes are more common.”

Related to this, in the following sentence the paper describes the shortcomings of SMD when the sample size is small. It would be useful to point out that Hedges' bias-corrected estimate is meant to address bias when sample sizes are small (Hedges, 1981).

We fully agree and have now incorporated the shortcoming in the estimation of the variance in Cohen’s *d* into the description of the difference between Cohen’s *d* and Hedges’ *g* (see response to reviewer 3, first comment).

In Section 1.2, the authors describe the relationship between funnel plot asymmetry and publication bias. It would also be worth pointing out that funnel plot asymmetry does not necessarily indicate publication bias. There are other reasons why funnel plots may appear asymmetrical, such as systematic heterogeneity related to the inclusion of two different modes of inquiry that differ both in effect magnitude and in typical standard error.

We agree and have therefore added an explanation of the fact that asymmetry may result from different sources to the Introduction.

Amendment(s) to the manuscript:

Section 1.2: “Importantly, there are other causes of asymmetry in funnel plots. For instance, the true effect size in smaller (and therefore less precise) studies may be genuinely different from that in large studies (for instance because the intensity of the intervention was higher in small studies). […] In addition, artefacts and chance may cause asymmetry (as shown e.g. in this study).”

There are two sections of the paper with the heading "Data Simulations" (Section 2.2 and Section 4.1). Results of the simulations are in Section 2.2. Section 4.1 is at the end of the paper and describes the process for simulating the data. This organization is confusing. It would be better if the description of the simulation were provided before these results, as it provides context for understanding the outcome of the author's simulation. Also, headings that clearly define these different sections will make it easier for readers to navigate the article.

Thank you for pointing this out. We followed *eLife*’s guidelines for authors when organizing the sections of the manuscript, which dictate that the Materials and methods section should come last, after the Results and Discussion. We have, however, changed the titles of the subheadings of Section 2.2 and 4.1 to avoid confusion of the sections.

Amendment(s) to the manuscript:

Section 2.2: changed section heading to “Data simulation results”

Section 4.2: changed section heading to “Data simulation methods”

In the third paragraph of Section 2.2 the authors state that results were similar for Cohen's d and Hedges' g effect-size estimates. Based on the information in the paper (Figure 3 legend and Figure 3—figure supplement 2), it appears that the comparisons for these estimates were only made using large sample sizes. In this case, similar results would be expected. They would be more likely to have different results in cases where sample sizes are small, as d and g are more-or-less identical for large sample sizes. The authors should address this scenario as well and make it clear that the information provided uses the Hedges’ g bias-corrected estimate of effect size.

We apologize for the confusion, the plots in Figure 4—figure supplement 1 are in fact of simulations with small sample sizes (n=12-30; 3000 studies). We have improved the figure legend to state this more clearly. In addition, we agree that it is worthwhile to present a more elaborate comparison of Hedges’ *g* with Cohen’s *d*, as also suggested by reviewer 1. We have therefore re-run all simulations in Cohen’s *d* and present the results of both methods in Supplementary file 1. The data presented further strengthen our conclusion that there are no apparent differences between the results of simulations using Hedges’ g versus those using Cohen’s *d*. To illustrate this more extensively, we have updated supplemental Figure 2 to include panels for all effect sizes (SMD=0, SMD=0.5 and SMD=1).

Amendment(s) to the manuscript:

Figure 4—Figure 2: added panels for SMD = 0 and SMD = 0.5, adjusted figure legend.

In the fourth paragraph of Section 2.2 the authors use SMD-1/√n to refer to their analysis of funnel plots looking at SMD with 1/√n on the y axis. The hyphen could be interpreted as arithmetic minus sign, so additional clarity of notation is needed.

Thank you for this helpful comment, which was also mentioned by reviewer #1. As mentioned above, we have replaced “SDM-”, “NMD-” and “RMD-” with “SMD vs.”, “NMD vs.” or “RMD vs.” throughout the manuscript, to avoid confusion of the hyphen with the minus sign.

In the fourth paragraph of Section 2.2 the authors state that the distortion is not seen in plot D of Figure 4. Visual interpretation of funnel plots is subjective. To some readers, the plot will continue to appear asymmetric, but flatter. The authors should explain what details of the plot lead them to the conclusion that the plot was symmetric as well as plots A and C, which are also described as not indicating distortion.

We agree that it is important to emphasize that visual inspection of funnel plots is subjective, and that the results of Egger’s regression test (reported in Table 2) were leading in our conclusions on funnel plot (a)symmetry in Figure 4. In addition, when inspecting the plots visually, the typical left-upward/right downward shift of the small and large SMD data points that we describe as typical for SMD versus SE funnel plots can be seen in Figure 4. We do not see this pattern in the other panels. However, we agree that these may still appear asymmetric or even be slightly asymmetric, since for all unbiased scenario’s, ~50 out of 1000 simulations were asymmetrical by chance.

Amendment(s) to the manuscript:

Section 2.2: “As in simulation 1, SMD vs. SE funnel plots of unbiased simulations were identified as asymmetrical by Egger’s test (Table 2). However, when the precision estimate was changed from SE to 1/√n, the prevalence of false positive results fell to the expected 5% (Table 2). For the NMD, Egger’s test performed correctly when using either the SE or 1/√n as precision estimate. In all scenario’s, approximately 50 out of 1000 simulated funnel plots appeared to be asymmetrical by chance (Table 2). The results of Egger’s test are supported by visual inspection of funnel plots of these unbiased scenario’s (Figure 5). The typical left-upward shift of the small SMD data points and right-downward shift of the large SMD data points is clearly visible in the SMD vs. SE plot (Figure 5), but not in the RMD, SMD vs. 1/√n or NMD plots.”

In Table 2, the authors of this study operationalize publication bias by removing all studies with a p-value ≥.10. This operation means that studies with a p-value above the cutoff have no chance of being published, and all studies with a p-value below (e.g. p=.09 and p=.01) have equal certainty of publication. This specification is not a good reflection of how publication bias functions in the real world. The authors should provide justification for using a simple cutoff of a p-value, rather than a model with diminishing probability such as a step function or a decaying continuous function. The authors should also include their reasoning for setting the cutoff at p ≥}.10, as well as information on how many studies were excluded based on this cutoff, such as the average number of studies remaining in the analysis.

We agree that our method to introduce publication bias can be refined and have therefore improved this analysis by introducing bias using a step wise function.

Amendment(s) to the manuscript:

Added description of step-wise method to section 4.2: “We then introduced publication bias in these meta-analyses using a stepwise method, Publication bias was introduced stepwise, by removing 10% of primary studies in which the difference between the intervention and control group means was significant at p<0.05 (Student-t test), 50% of studies where the significance level was p≥0.05 to p<0.10, and 90% of studies where the significance level was p≥0.10.”

Added R script to Supplementary file 1.

Supplemental figure now shows example plots of the new step function.

In Section 2.3, the authors compare the results of the analyses using examples from real-world studies. Providing this shows how their research may be applicable in real world settings. It would also be useful to report effect-size estimate from the original meta-analysis.

We agree that this would indeed be helpful to the reader, and have therefore added the original and adjusted effect sizes of the meta-analyses in Table 3 (see also reviewer 1, comment 3, second point).

The Discussion section is missing a discussion of the limitations of the current study. The simulation had a limited number of conditions. Providing limitations identifies the scope of the presented findings.

We have added the limitations of our data simulations to section 3.1 Limitations.

Amendment(s) to the manuscript:

Section 3.1: “We designed our data simulations to closely resemble empirical data in terms of the range of sample sizes, effect sizes and numbers of studies in a meta-analyses. […] However, research on how to optimally simulate these parameters is first needed, and was beyond the scope of this study. Instead, we used re-analyses of empirical data to test our proposed solutions on a number of real-life meta-analyses which include all of these aspects.”

Also, the authors should include a rationale explaining why they got these results Without that, the findings atheoretical; entirely empirical methodological findings are harder to accept with confidence.

We have added Section 1.3 and a new (now Figure 2) in which we discuss the theoretical rationale of SMD vs. SE funnel plot distortion in more detail. We aim for the theoretical explanation to be suitable for the level of statistical knowledge of our target audience (review authors/ readers of reviews/ reviewers of reviews).

Amendment(s) to the manuscript:

Added section 1.3 Theoretical explanation of SMD funnel plot distortion:

“1.3 Theoretical explanation of SMD funnel plot distortion

In a meta-analysis using the SMD as effect measure, in the absence of publication bias, observed SMDs in a funnel plot will be scattered around the true underlying SMD. *[…]* The effect of the second component on the SE, and the resulting distortion, is largest if the sample size is small and the SMD is large (Figure 2).”Added Figure 2, visualisation of the theoretical explanation.

In the first paragraph of Section 4.1, the authors describe the process for obtaining individual study sample sizes in the simulation. They should provide a rationale for sampling study sample sizes from a uniform distribution. Sample sizes typically do not follow a uniform distribution; rather they tend to be positively skewed. This brings into question the relevance of the simulations.

As described on in our tenth response to your major comments, we have added a description to Section 3.1 of the limitations of or simulation approach that closely resembles the sample sizes, effect sizes and number of studies of empirical data, but not varying degrees of heterogeneity, unequal variances between groups or sampling from non-normally distributed data. This we feel is beyond the scope of this study.

In Table 4 legend the authors describe the magnitudes for the effect sizes included in the simulation. They should provide a reasoning for using an effect magnitude of SMD=1. By many standards, this would be considered a very large effect.

This question was also raised by reviewer 1. We kindly refer you to our response to reviewer 1, major comment 2, first point, for an explanation and amendments to the manuscript.

It is possible that the method would perform better (or worse) under conditions where the population magnitude is small or moderate.

We interpreted ‘population magnitude’ as the effect size introduced. Although this method might indeed reach significance later, the methodological problem still exists. Slightly more power may be needed to detect it, but chances of false positive claims are still higher than they should be.

Also, the simulation did not include heterogeneity of studies in the model: all studies were sampled from exactly the same fixed-effect distribution. Studies conducted in the real world tend to have heterogeneity. Conditions of heterogeneity often lead to failure of trim and fill and Eggers regression, and would influence the outcome of an analysis of publication bias.

We fully agree that heterogeneity in “real” meta-analyses may greatly influence the outcomes of Egger’s regression and trim and fill analyses. However, the goal of our current simulations is to illustrate the cause of the problem (why are SMD vs. SE funnel plots distorted), and how is this influenced by three straightforward variables: sample size, effect size and number of studies in the analysis. We feel that introducing heterogeneity into these simulations will not make the message more clear. Because heterogeneity is such a complex phenomenon, we felt it was more informative to perform re-analyses of empirical datasets, in which heterogeneity and the aforementioned variables are combined and which therefore optimally serve the purpose of showing how relevant our findings are in practice.